# Demystifying Orthogonal Monte Carlo and Beyond

**Han Lin** *
Columbia University
hl3199@columbia.edu

**Haoxian Chen** *
Columbia University
hc3136@columbia.edu

**Tianyi Zhang**
Columbia University
tz2376@columbia.edu

**Clement Laroche**
Columbia University
cl3778@columbia.edu

**Krzysztof Choromanski**
Google Brain Robotics & Columbia University
kchoro@google.com

## Abstract

Orthogonal Monte Carlo [43] (OMC) is a very effective sampling algorithm imposing structural geometric conditions (orthogonality) on samples for variance reduction. Due to its simplicity and superior performance as compared to its Quasi Monte Carlo counterparts, OMC is used in a wide spectrum of challenging machine learning applications ranging from scalable kernel methods [18] to predictive recurrent neural networks [11], generative models [36] and reinforcement learning [16]. However theoretical understanding of the method remains very limited. In this paper we shed new light on the theoretical principles behind OMC, applying theory of negatively dependent random variables to obtain several new concentration results. As a corollary, we manage to obtain first uniform convergence results for OMCs and consequently, substantially strengthen best known downstream guarantees for kernel ridge regression via OMCs. We also propose novel extensions of the method leveraging theory of algebraic varieties over finite fields and particle algorithms, called *Near-Orthogonal Monte Carlo* (NOMC). We show that NOMC is the first algorithm consistently outperforming OMC in applications ranging from kernel methods to approximating distances in probabilistic metric spaces.

## 1 Introduction & Related Work

Monte Carlo (MC) methods are widely applied in machine learning in such domains as: dimensionality reduction [1, 3, 2], scalable kernel methods with random feature maps [34], generative modeling and variational autoencoders via sliced Wasserstein distances [36], approximating Gaussian smoothings in Evolutionary Strategies (ES) algorithms for Reinforcement Learning (RL) [16], predictive recurrent neural networks [11] and more. The theory of MC is rich with various techniques improving the accuracy of base MC estimators such as: antithetic couplings and importance sampling [6], variance reduction via carefully designed control variate terms [29, 33] and finally: the vast field of the so-called *Quasi Monte Carlo* (QMC) methods [7, 27, 20, 19].

Relatively recently, yet another algorithm which can be combined with most of the aforementioned approaches, called *Orthogonal Monte Carlo* (OMC) has been proposed [43]. OMC relies on ensembles of mutually orthogonal random samples for variance reduction and turns out to be very effective in virtually all applications of MC in machine learning involving isotropic distributions [36, 11, 18, 14, 35]. Providing substantial accuracy improvements over MC baselines, conceptually simple, and superior to algorithms leveraging QMC techniques, it became one of the most frequently used techniques in a vast arsenal of MC tools.

OMCs are also much simpler than the class of MC methods based on determinantal point processes (DPPs) [28]. DPPs provide elegant mechanisms for sampling diverse ensembles, where diversity is encoded by a kernel. Some DPP-MCs [23, 12, 8] provide stronger theoretical guarantees than base

---

MCs, yet those are impractical to use in higher dimensions due to their prohibitive time complexity, especially when samples need to be frequently constructed such as in RL [16].

Despite its effectiveness and impact across the field, theoretical principles behind the OMC method remain only partially understood, with theoretical guarantees heavily customized to specific applications and hard to generalize to other settings [35, 14, 17].

In this paper we shed new light on the effectiveness of OMCs by applying theory of *negatively dependent* random variables that is a theoretical backbone of DPPs. Consequently, we present first comprehensive theoretical view on OMCs. Among our new results are first exponentially small probability bounds for errors of OMCs applied to objectives involving **general nonlinear mappings**. Previously such results were known only for the cosine mapping in the setting of Gaussian kernel approximation via random features [11] and for random linear projections for dimensionality reduction. Understanding the effectiveness of OMCs in the general nonlinear setting was considered the Holy Grail of the research on structured MC methods, with elusive general theory. This striking discrepancy between practice where OMCs are used on a regular basis in general nonlinear settings and very limited developed theory is one of the main motivations of this work.

Our techniques enable us to settle several open conjectures for OMCs. Those involve not only aforementioned results for the general nonlinear case, but strong concentration results for arbitrary RBF kernels with no additional assumptions regarding corresponding spectral densities, in particular first such results for all Matérn kernels. We show that our concentration results directly imply uniform convergence of OMCs (which was an open question) and that these lead to substantial strengthening of the best known results for kernel ridge regression via OMCs from [11]. The strengthenings are twofold: we extend the scope to all RBF kernels as opposed to just *smooth RBFs* [11] and we significantly improve accuracy guarantees.

One of the weaknesses of OMCs is that orthogonal ensembles can be defined only if the number of samples $s$ satisfies $s \leq d$, where $d$ stands for data dimensionality. In such a setting a relaxed version of the method is applied, where one orthogonal block is replaced by multiple independent orthogonal blocks [43]. Even though orthogonal entanglement of samples across different blocks is now broken, such block-orthogonal OMC methods (or B-OMCs) were still the most accurate known MC algorithms for isotropic distributions when $s \gg d$.

We propose an extension of OMCs relying on the ensembles of random near-orthogonal vectors preserving entanglements across all the samples, called by us *Near-Orthogonal Monte Carlo* (NOMC), that to the best of our knowledge, is the first algorithm beating B-OMCs. Previously, related ideas are discussed in [37, 10, 30]. [37, 10] summarize different types of energy formulations and algorithms for distributing points over the sphere, showing that minimizing the discrete Riesz energy over the sphere $\mathcal{S}^{d-1}$ can produce asymptotically uniformly distributed points. Based on this, [30] proposes a coordinate descent algorithm to find a local minimum of the discrete Riesz energy and applies to kernel approximation setting, especially in Gaussian kernel and $b$-th order arc-cosine kernels. Motivated by these, we propose an entirely new energy formulation of points over sphere, and propose a simpler optimization algorithm. In addition, we demonstrate it in different settings such as: kernel approximation methods (beyond Gaussian and $b$-th order arc-cosine kernels) and approximating *sliced Wasserstein distances* (used on a regular basis in generative modeling). NOMCs are based on two new paradigms for constructing structured MC samples: high-dimensional optimization with particle methods and the theory of algebraic varieties over finite fields.

We highlight our main contributions below.

- By leveraging the theory of negatively dependent random variables, we provide first exponentially small bounds on error probabilities for OMCs used to approximate objectives involving **general nonlinear mappings** [Sec. 3; Theorem 1, Theorem 2].

- We show how our general theory can be used to obtain simpler proofs of several known results and new results not known before [Sec. 2, Sec. 3.1], e.g. first Chernoff-like concentration inequalities regarding certain classes of Pointwise Nonlinear Gaussian (PNG) kernels and all RBF kernels (previously such results were known only for RBF kernels with corresponding isotropic distributions of no heavy tails [15, 11]).

- Consequently, we provide first uniform convergence results for OMCs and as a corollary, apply them to obtain new SOTA downstream guarantees for kernel ridge regression with OMCs [Sec. 3.1.1], improving both: accuracy and scope of applicability.

- We propose two new paradigms for constructing structured samples for MC methods when $s \gg d$, leveraging number theory techniques and particle methods for high-dimensional optimization. In particular, we apply a celebrated Weil Theorem [41] regarding generating functions derived from counting the number of points on algebraic varieties over finite fields.

- We empirically demonstrate the effectiveness of NOMCs [Sec. 5].

## 2 Orthogonal Monte Carlo

Consider a measurable function $f_{\mathcal{Z}} : \mathbb{R}^d \to \mathbb{R}^k$ parameterized by an ordered subset $\mathcal{Z} \subseteq_{\mathrm{ord}} \mathbb{R}^d$ with:

$$F_{f,\mathcal{D}}(\mathcal{Z}) \overset{\text{def}}{=} \mathbb{E}_{\omega \sim \mathcal{D}}[f_{\mathcal{Z}}(\omega)], \qquad (1)$$

where $\mathcal{D}$ is an isotropic probability distribution on $\mathbb{R}^d$ (isotropic distribution in this paper is defined as having constant probabilistic density function on every sphere centered at zero). In this work we analyze MC-based approximation of $F$. Examples of important machine learning instantiations of $F$ are given below.

**Kernel Functions & Random Features:** Every shift-invariant kernel $K : \mathbb{R}^d \times \mathbb{R}^d \to \mathbb{R}$ can be written as $K(\mathbf{x}, \mathbf{y}) = g(\mathbf{x} - \mathbf{y}) \overset{\text{def}}{=} \mathbb{E}_{\omega \sim \mathcal{D}}[\cos(\omega^\top (\mathbf{x} - \mathbf{y}))]$ for some probability distribution $\mathcal{D}$ [34]. Furthermore, if $K$ is a *radial basis function* (RBF) kernel (e.g. Gaussian or Matérn), i.e. $K(\mathbf{x}, \mathbf{y}) = r(\|\mathbf{x} - \mathbf{y}\|_2)$ for some $r : \mathbb{R}_{\geq 0} \to \mathbb{R}$, then $\mathcal{D}$ is isotropic. Here $\mathcal{Z} = (\mathbf{z})$, where $\mathbf{z} = \mathbf{x} - \mathbf{y}$, and $f_{(\mathbf{z})}(\omega) = \cos(\omega^\top \mathbf{z})$. For *pointwise nonlinear Gaussian* [PNG] kernels [18] (e.g. angular or arc-cosine), given as $K_h(\mathbf{x}, \mathbf{y}) = \mathbb{E}_{\omega \sim \mathcal{N}(0, \mathbf{I}_d)}[h(\omega^\top \mathbf{x})h(\omega^\top \mathbf{y})]$, where $h : \mathbb{R} \to \mathbb{R}$, the corresponding distribution $\mathcal{D}$ is multivariate Gaussian and $f$ is given as $f_{(\mathbf{x}, \mathbf{y})} = h(\omega^\top \mathbf{x})h(\omega^\top \mathbf{y})$.

**Dimensionality Reduction [JLT]:** Johnson-Lindenstrauss dimensionality reduction techniques (JLT) [26, 1, 31] rely on embeddings of high-dimensional feature vectors via random projections given by vectors $\omega \sim \mathcal{N}(0, \mathbf{I}_d)$. Expected squared distances between such embeddings of input high-dimensional vectors $\mathbf{x}, \mathbf{y} \in \mathbb{R}^d$ are given as: $\mathrm{dist}_{\mathrm{JLT}}^2(\mathbf{x}, \mathbf{y}) = \mathbb{E}_{\omega \sim \mathcal{N}(0, \mathbf{I}_d)}[(\omega^\top (\mathbf{x} - \mathbf{y}))^2]$. Here $\mathcal{D}$ is multivariate Gaussian and $f_{(\mathbf{z})} = (\omega^\top \mathbf{z})^2$ for $\mathbf{z} = \mathbf{x} - \mathbf{y}$.

**Sliced Wasserstein Distances [SWD]:** Wasserstein Distances (WDs) are metrics in spaces of probability distributions that have found several applications in deep generative models [5, 24]. For $p \geq 1$, the $p$-th Wasserstein distance between two distributions $\eta$ and $\mu$ over $\mathbb{R}^d$ is defined as:

$$\mathrm{WD}_p(\eta, \mu) = \left( \inf_{\gamma \in \Gamma(\eta, \mu)} \int_{\mathbb{R}^d \times \mathbb{R}^d} \|\mathbf{x} - \mathbf{y}\|_2^p d\gamma(\mathbf{x}, \mathbf{y}) \right)^{\frac{1}{p}},$$

where $\Gamma(\eta, \mu)$ is the set of joint distributions over $\mathbb{R}^d \times \mathbb{R}^d$ for which the marginal of the first/last $d$ coordinates is $\eta/\mu$. Since WD computations involve solving nontrivial optimal transport problem (OPT) [38] in the high-dimensional space, in practice its more efficient to compute proxies are used, among them the so-called *Sliced Wasserstein Distance* (SWD) [9]. SWDs are obtained by constructing projections $\eta_{\mathbf{u}}$ and $\mu_{\mathbf{u}}$ of $\eta$ and $\mu$ into a random 1d-subspace encoded by $\mathbf{u} \sim \mathrm{Unif}(\mathcal{S}^{d-1})$ chosen uniformly at random from the unit sphere $\mathcal{S}^{d-1}$ in $\mathbb{R}^d$ (see: Sec. 5). If $\eta$ and $\mu$ are given as point clouds, they can be rewritten as in Eq. 1, where $\mathcal{Z}$ encodes $\eta$ and $\mu$ via cloud points.

### 2.1 Structured ensembles for Monte Carlo approximation

A naive way of estimating function $F_{f,\mathcal{D}}(\mathcal{Z})$ from Eq. 1 is to generate $s$ independent samples : $\omega_1^{\mathrm{iid}}, ..., \omega_s^{\mathrm{iid}} \overset{\text{iid}}{\sim} \mathcal{D}$, which leads to the base unbiased Monte Carlo (MC) estimator:

$$\widehat{F}_{f,\mathcal{D}}^{\mathrm{iid}}(\mathcal{Z}) \overset{\text{def}}{=} \frac{1}{s} \sum_{i=1}^{s} f_{\mathcal{Z}}(\omega_i^{\mathrm{iid}}). \qquad (2)$$

Orthogonal Monte Carlo (OMC) method relies on the isotropicity of $\mathcal{D}$ and instead entangles different samples in such a way that they are **exactly orthogonal**, while their marginal distributions match those of $\omega_i^{\mathrm{iid}}$ (this can be easily done for instance via Gram-Schmidt orthogonalization followed by row-renormalization, see: [43]). Such an ensemble $\{\omega_1^{\mathrm{ort}}, ..., \omega_s^{\mathrm{ort}}\}$ is then used to replace $\{\omega_1^{\mathrm{iid}}, ..., \omega_s^{\mathrm{iid}}\}$ in Eq. 2 to get OMC estimator $\widehat{F}_{f,\mathcal{D}}^{\mathrm{ort}}(\mathcal{Z})$.

Estimator $\widehat{F}_{f,\mathcal{D}}^{\mathrm{ort}}(\mathcal{Z})$ can be constructed only if $s \leq d$, where $d$ stands for samples' dimensionality. In most practical applications we have: $s > d$ and thus instead the so-called *block orthogonal Monte*

*Carlo* (B-OMC) procedure is used, where $s$ samples are partitioned into $d$-size blocks, samples within each block are chosen as above and different blocks are constructed independently [43]. In B-OMC, orthogonality is preserved locally within a block, but this entanglement is lost across the blocks.

In the next section we provide new general theoretical results for OMCs.

## 3 Orthogonal Monte Carlo and Negatively Dependent Ensembles

For a rigorous analysis, we will consider an instantiation of the objective from Eq. 1 of the form:

$$F_{f,\mathcal{D}}(\mathbf{z}) = \mathbb{E}_{\omega \sim \mathcal{D}}[f(\omega^\top \mathbf{z})] \tag{3}$$

for $\mathbf{z} \in \mathbb{R}^d$ and some measurable function $f : \mathbb{R} \to \mathbb{R}$. We consider classes of functions $f(u)$ satisfy:

- **F1.** monotone increasing or decreasing in $|u|$,
- **F2.** decomposable as $f = f^+ + f^-$, where $f^+$ is monotone increasing and $f^-$ is monotone decreasing in $|u|$,
- **F3.** entire (i.e. expressible as Taylor series with infinite radius of convergence, e.g. polynomials).

**Remark:** As we will see later, for the class **F3** the role of $f^+$ and $f^-$ in the analysis is taken by functions: $\mathrm{even}[f]^+$ and $\mathrm{even}[f]^-$, where $\mathrm{even}[f]$ stands for function obtained from $f$ by taking terms of the Taylor series expansion corresponding to even powers.

Such objectives $F_{f,\mathcal{D}}$ are general enough to cover: dimensionality reduction setting, all RBF kernels, certain classes of PNG kernels and several statistics regarding neural network with random weights (see: Sec. 3.1) that we mentioned before. See also Table 1, where we give an overview of specific examples of functions covered by us, and Sec. 3.1 for much more detailed analysis of applications.

For a random variable $X$ we define moment generating function $M_X$ as: $M_X(\theta) = \mathbb{E}[e^{\theta X}]$. Furthermore, we define *Legendre transform* as: $\mathcal{L}_X(a) = \sup_{\theta>0} \log(\frac{e^{\theta a}}{M_X(\theta)})$ if $a > \mathbb{E}[X]$ and $\mathcal{L}_X(a) = \sup_{\theta<0} \log(\frac{e^{\theta a}}{M_X(\theta)})$ if $a < \mathbb{E}[X]$. It is a standard fact from probability theory that $\mathcal{L}_X(a) > 0$ for every $a \neq \mathbb{E}[X]$.

We prove first exponentially small bounds for failure probabilities of OMCs applied to functions from all three classes and in addition show that for the class **F1** obtained concentration bounds are better than for the base MC estimator.

Our results can be straightforwardly extended to classes of functions expressible as limits of functions from the above **F1**-**F3**, but for the clarity of the exposition we skip this analysis. To the best of our knowledge, we are the first to provide theory that addresses also **discontinuous** functions.

| | JLT | PNG: $h(x) = e^{cx}$ | Gaussian | $\nu$-Matérn |
|---|---|---|---|---|
| class | **F1,F3** | **F3** | **F2,F3** | **F2,F3** |
| $f^+(u)/\mathrm{even}[f]^+(u)$ | $x^2$ | $\sum_{k=0}^{\infty} \frac{(cu)^{2k}}{(2k)!}$ | $\sum_{k=0}^{\infty} \frac{u^{4k}}{(4k)!}$ | $\sum_{k=0}^{\infty} \frac{u^{4k}}{(4k)!}$ |
| $f^-(u)/\mathrm{even}[f]^-(u)$ | N/A | N/A | $-\sum_{k=0}^{\infty} \frac{u^{4k+2}}{(4k+2)!}$ | $-\sum_{k=0}^{\infty} \frac{u^{4k+2}}{(4k+2)!}$ |
| SOTA results for OMC | ortho-JLTs [26] | **ours** | [11] | **ours**: any $\nu$ |

Table 1: Examples of particular instantiations of function classes **F1**-**F3** covered by our theoretical results.

The key tool we apply to obtain our general theoretical results is the notion of *negative dependence* [ND] [32, 40] that is also used in the theory of Determinantal Point Processes (DPPs) [28]:

**Definition 1** (Negative Dependence (ND)). *Random variables $X_1, \dots, X_n$ are said to be negatively dependent if both of the following two inequalities hold for any $x_1, \dots, x_n \in \mathbb{R}$*

$$\mathbb{P}(\bigcap_i X_i \geq x_i) \leq \prod_i \mathbb{P}(X_i \geq x_i), \text{ and } \mathbb{P}(\bigcap_i X_i \leq x_i) \leq \prod_i \mathbb{P}(X_i \leq x_i).$$

We show that certain classes of random variables built on orthogonal ensembles satisfy ND property:

**Lemma 1** (ND for OMC-samples and monotone functions). *For an isotropic distribution $\mathcal{D}$ on $\mathbb{R}^d$ and orthogonal ensemble: $\omega_1^{\mathrm{ort}}, \dots, \omega_d^{\mathrm{ort}}$ with $\omega_i^{\mathrm{ort}} \sim \mathcal{D}$, random variables: $X_1, \dots, X_d$ defined as: $X_i = |\omega^{\mathrm{ort}}{}_i^\top \mathbf{z}|$ are negatively dependent for any fixed $\mathbf{z} \in \mathbb{R}^d$.*

Lemma 1 itself does not guarantee strong convergence for orthogonal ensembles however is one of the key technical ingredients that helps us to achieve this goal. The following is true:

**Lemma 2.** *Assume that $f$ is a function from the class **F1**. Let $X_i = f(\omega_i^{\mathrm{ort}\top} \mathbf{z})$ for $i = 1, ..., n$, and let $\lambda$ be a non-positive (or non-negative) real number. Then the following holds:*

$$\mathbb{E}[e^{\lambda \sum_{i=1}^m X_i}] \leq \prod_{i=1}^m \mathbb{E}[e^{\lambda X_i}].$$

Note that Lemma 2 lead directly to the following corollary relating iid and orthogonal ensembles:

**Corollary 1** (exponentials of OMCs and MCs). *Let $\mathbf{z} \in \mathbb{R}^d$ and assume that function $f : \mathbb{R} \to \mathbb{R}$ is from the class **F1**. Take an isotropic distribution $\mathcal{D}$ on $\mathbb{R}^d$, an ensemble of independent samples $\omega_1^{\mathrm{iid}}, ..., \omega_s^{\mathrm{iid}}$ and an orthogonal ensemble $\omega_1^{\mathrm{ort}}, ..., \omega_s^{\mathrm{ort}}$ giving rise to base MC estimator $\widehat{F}_{f,\mathcal{D}}^{\mathrm{iid}}(\mathbf{z})$ of $\mathbb{E}_{\omega \sim \mathcal{D}}[f(\omega^\top \mathbf{z})]$ and to its orthogonal version $\widehat{F}_{f,\mathcal{D}}^{\mathrm{ort}}(\mathbf{z})$. Then the following is true for any $\lambda$:*

$$\mathbb{E}[e^{\lambda \widehat{F}_{f,\mathcal{D}}^{\mathrm{ort}}(\mathbf{z})}] \leq \mathbb{E}[e^{\lambda \widehat{F}_{f,\mathcal{D}}^{\mathrm{iid}}(\mathbf{z})}]. \tag{4}$$

Corollary 1 enables us to obtain stronger concentrations results for OMCs than for base MCs for the class **F1**. By combining it with extended Markov's inequality, we obtain the following:

**Theorem 1** (OMC-bounds surpassing MC-bounds for the **F1**-class). *Denote by $\mathrm{MSE}$ a mean squared error of the estimator, by $s$ the number of MC samples used and let $X = f(\omega^\top \mathbf{z})$ for $\omega \sim \mathcal{D}$. Then under assumptions as in Corollary 1, OMC leads to the unbiased estimator satisfying for $\epsilon > 0$:*

$$\mathbb{P}[|\widehat{F}_{f,\mathcal{D}}^{\mathrm{ort}}(\mathbf{z}) - F_{f,\mathcal{D}}(\mathbf{z})| \geq \epsilon] \leq p(\epsilon), \tag{5}$$

*where $p(\epsilon)$ is defined as: $p(\epsilon) \stackrel{\mathrm{def}}{=} \exp(-s\mathcal{L}_X(F_{f,\mathcal{D}}(\mathbf{z})+\epsilon) + \exp(-s\mathcal{L}_X(F_{f,\mathcal{D}}(\mathbf{z})-\epsilon)$ for unbounded $f$, and is defined as $p(\epsilon) \stackrel{\mathrm{def}}{=} 2\exp(-\frac{2s\epsilon^2}{(b-a)^2})$ for bounded $f \in [a,b]$, which is a standard upper bound on $\mathbb{P}[|\widehat{F}_{f,\mathcal{D}}^{\mathrm{iid}}(\mathbf{z}) - F_{f,\mathcal{D}}(\mathbf{z})| \geq \epsilon]$. Furthermore: $\mathrm{MSE}(\widehat{F}_{f,\mathcal{D}}^{\mathrm{ort}}(\mathbf{z})) \leq \mathrm{MSE}(\widehat{F}_{f,\mathcal{D}}^{\mathrm{iid}}(\mathbf{z}))$.*

For bounded **F1**-class, $a, b$ are easy to determine. For instance, in the kernel approximation setting, most of the kernels are bounded, implying that their non-linear mapping $f$ must be bounded as well.

For functions from **F2**-class, we simply decompose $f$ into its monotone increasing ($f^+$) and decreasing ($f^-$) part, apply introduced tools independently to $f^+$ and $f^-$ and use union bound. Finally, if $f$ is taken from the **F3**-class, we first decompose it into $\mathrm{even}[f]$ and $\mathrm{odd}[f]$ components, by leaving only even/odd terms in the Taylor series expansion. We then observe that for isotropic distributions we have: $F_{\mathrm{odd}[f],\mathcal{D}} = 0$ (see: Appendix Sec. 9.4) and thus reduce the analysis to that of $\mathrm{even}[f]$ which is from the **F2**-class. We conclude that:

**Theorem 2** (Exponential bounds for OMCs and **F2**/**F3** classes). *Let $\mathbf{z} \in \mathbb{R}^d$ and assume that function $f : \mathbb{R} \to \mathbb{R}$ is from the class **F2** or **F3**. Then for $\epsilon > 0$:*

$$\mathbb{P}[|\widehat{F}_{f,\mathcal{D}}^{\mathrm{ort}}(\mathbf{z}) - F_{f,\mathcal{D}}(\mathbf{z})| \geq \epsilon] \leq p(\epsilon) \stackrel{\mathrm{def}}{=} u^+ + u^-, \tag{6}$$

*where $u^{+/-} \stackrel{\mathrm{def}}{=} \exp(-s\mathcal{L}_{X^{+/-}}(F_{f,\mathcal{D}}(\mathbf{z})+\frac{\epsilon}{2}) + \exp(-s\mathcal{L}_{X^{+/-}}(F_{f,\mathcal{D}}(\mathbf{z})-\frac{\epsilon}{2})$, and $X^{+/-}$ is defined as: $X^{+/-} \stackrel{\mathrm{def}}{=} f^{+/-}$ if $f$ is from **F2** and as: $X^{+/-} \stackrel{\mathrm{def}}{=} (\mathrm{even}[f])^{+/-}$ if $f$ is from **F3**. As before, in the bounded case we can simplify $u^+$ and $u^-$ to: $u^{+/-} \stackrel{\mathrm{def}}{=} 2\exp(-\frac{s\epsilon^2}{2(b^{+/-}-a^{+/-})^2})$, where $a^+, b^+, a^-, b^-$ are lower and upper bounds such that: $f^+ \in [a^+, b^+]$ and $f^- \in [a^-, b^-]$ if $f$ is from **F2** or $(\mathrm{even}[f])^+ \in [a^+, b^+]$ and $(\mathrm{even}[f])^- \in [a^-, b^-]$ if $f$ is from **F3**. Furthermore, if $(\mathrm{even}[f])^+ = 0$ or $(\mathrm{even}[f])^- = 0$, we can tighten that bound using upper bound from Theorem 1 and thus, establish better concentration bounds than for base MC.*

The proofs of all our theoretical results are given in the Appendix.

## 3.1 Applications

In this section we discuss in more detail applications of the presented results. We see that by taking $f(x) = x^2$, we can apply our results to the JLT setting from Sec. 2.

**General RBF kernels:** Even more interestingly, Theorem 2 enables us to give first strong concentration results for all RBF kernels, avoiding very cumbersome technical requirements regarding tails of the corresponding spectral distributions (see: [15, 11]). In particular, we affirmatively answer an open question whether OMCs provide exponential concentration guarantees for the class of Matérn kernels for every value of the hyperparameter $\nu$ ([15]). Theorem 2 can be also directly applied to obtain strong concentration results regarding kernel ridge regression with OMCs (see: Theorem 2 from [11]) for all RBF kernels as opposed to just *smooth RBFs* as in [11]. Thus we bridge the gap between theory (previously valid mainly for Gaussian kernels) and practice (where improvements via OMCs were reported for RBF kernels across the board [15]).

**First Strong Results for Classes of PNG Kernels:** We also obtain first exponentially small upper bounds on errors for OMCs applied to PNG kernels, which were previously intractable and for which the best known results were coming from second moment methods [18, 13]. To see this, note that PNGs defined by nonlinearity $h(x) = e^{cx}$ can be rewritten as functions from the class **F3** (with $\mathbf{z} = \mathbf{x} + \mathbf{y}$), namely: $K_h(\mathbf{x}, \mathbf{y}) = \mathbb{E}_{\omega \sim \mathcal{N}(0, \mathbf{I}_d)}[e^{c\omega^\top(\mathbf{x}+\mathbf{y})}]$ (see: Table 1). Furthermore, by applying Theorem 2, we actually show that these bounds are better than for the base MC estimator.

### 3.1.1 Uniform Convergence for OMCs and New Kernel Ridge Regression Results

Undoubtedly, one of the most important applications of results from Sec. 3 are first uniform convergence guarantees for OMCs which provide a gateway to strong downstream guarantees for them, as we will show on the example of kernel ridge regression. MSE-results for OMCs from previous works suffice to provide some pointwise convergence, but are too weak for the uniform convergence and thus previous downstream theoretical guarantees for OMCs were not practical. The following is true and implies in particular that OMCs converge uniformly for all RBF kernels :

**Theorem 3** (Uniform convergence for OMCs). *Let $\mathcal{M} \subseteq \mathbb{R}^d$ be compact with diameter $\mathrm{diam}(\mathcal{M})$. Assume that $f$ has Lipschitz constant $L_f$. Then under assumptions as in Theorem 1 / 2, for any $r > 0$:*

$$\mathbb{P}[\sup_{\mathbf{z} \in \mathcal{M}} |\widehat{F}^{\mathrm{ort}}_{f,\mathcal{D}}(\mathbf{z}) - F_{f,\mathcal{D}}(\mathbf{z})| \geq \epsilon] \leq C(\frac{\mathrm{diam}(\mathcal{M})}{r})^d \cdot p(\epsilon/2) + (\frac{2r\sigma L_f}{\epsilon})^2, \tag{7}$$

*where $\sigma^2 = \mathbb{E}_{\omega \sim D}[\omega^T \omega]$ (i.e. the second moment of D), p is as in RHS of inequality from Theorem 1 / 2 and $C > 0$ is a universal constant. In particular, if boundedness conditions from Theorem 1 / 2 are satisfied, one can take: $s = \Theta(\frac{d}{\epsilon^2} \log(\frac{\sigma L_f \mathrm{diam}(\mathcal{M})}{\epsilon}))$ to get uniform $\epsilon$-error approximation with any constant probability (e.g $s = \Theta(\frac{d}{\epsilon^2} \log(\frac{d L_f \mathrm{diam}(\mathcal{M})}{\epsilon}))$ for Gaussian kernel for which $\sigma = d$).*

We can directly apply these results to kernel ridge regression with an **arbitrary** RBF via OMCs, by noting that in the RHS of Theorem 2 from [11] upper-bounding the error, we can drop $N^2$ multiplicative factor if all points are in the compact set (see: Appendix Sec. 9.7). This term was added as a consequence of simple union bound, no longer necessary if uniform convergence is satisfied.

## 4 Near-Orthogonal Monte Carlo Algorithm

Near-Orthogonal Monte Carlo (or: NOMC) is a new paradigm for constructing entangled MC samples for estimators involving isotropic distributions if the number of samples required satisfies $s > d$. We construct NOMC-samples to make points as evenly distributed on the unit sphere as possible. The angles between any two samples will be close to orthogonal when d is large (note that they cannot be exactly orthogonal for $s > d$ since this would imply their linear independence). That makes their distribution much more uniform than in other methods (see: Fig. 1).

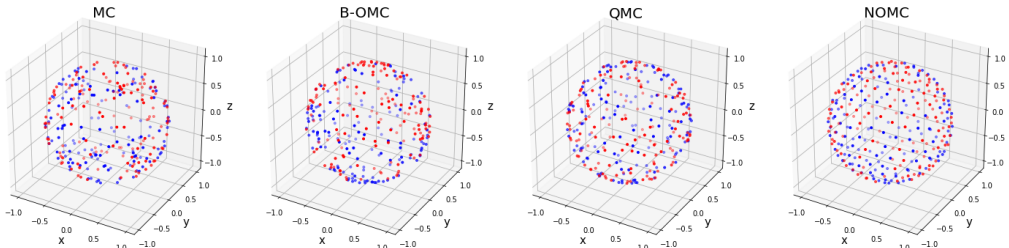

Figure 1: Visual comparison of the distribution of samples for four methods for $d = 3$ and $s = 150$. From left to right: base MC, B-OMC, QMC using Halton sequences and our NOMC. The color of points represent the direction of sampled vectors, with red for head and blue for tail. We see that NOMC produces most uniformly distributed samples.

This has crucial positive impact on the accuracy of the estimators applying NOMCs, making them superior to other methods, as we demonstrate in Sec. 5, and still unbiased.

There are two main strategies that we apply to obtain near-orthogonality surpassing this in QMC or B-OMC. Our first proposition is to cast sample-construction as an optimization problem, where near-orthogonal entanglement is a result of optimizing objectives involving angles between samples. We call this approach: opt-NOMC. Even though such an optimization incurs only one-time additional cost, we also present alg-NOMC algorithm that has lower time complexity and is based on the theory of algebraic varieties over finite fields. Algorithm alg-NOMC does not require optimization and in

practice gives similar accuracy, thus in the experimental section we refer to both simply as NOMC. Below we discuss both strategies in more detail.

## 4.1 Algorithm opt-NOMC

The idea of Algorithm opt-NOMC is to force repelling property of the samples/particles (that for the one-block case was guaranteed by the ND-property) via specially designed energy function.

That energy function achieves lower values for well-spread samples/particles and can be rewritten as the sum over energies $E(\omega_i, \omega_j)$ of local particle-particle interactions. There are many good candidates for $E(\omega_i, \omega_j)$. We chose: $E(\omega_i, \omega_j) = \frac{\delta}{\delta + \|\omega_i - \omega_j\|_2^2}$, where $\delta > 0$ is a tunable hyper-parameter. We minimize such an energy function on the sphere using standard gradient descent approach with projections. Without loss of generality, we can assume that the isotropic distribution $\mathcal{D}$ under consideration is a uniform distribution on the sphere $\text{Unif}(\mathcal{S}^{d-1})$, since for other isotropic distributions we will only need to conduct later cheap renormalization of samples' lengths. When the optimization is completed, we return randomly rotated ensemble, where random rotation is encoded by Gaussian orthogonal matrix obtained via standard Gram-Schmidt orthogonalization of the Gaussian unstructured matrix (see: [43]). Random rotations enable us to obtain correct marginal distributions (while keeping the entanglement of different samples obtained via optimization) and consequently - unbiased estimators when such ensembles are applied. We initialize optimization with an ensemble corresponding to B-OMC as a good quality starting point. For the pseudocode of opt-NOMC, see Algorithm 1 box.

**Remark:** Note that even though in higher-dimensional settings, such an optimization is more expensive, this is a **one time cost**. If new random ensemble is needed, it suffices to apply new random rotation on the already optimized ensemble. Applying such a random rotation is much cheaper and can be further sped up by using proxies of random rotations (see: [18]). The time complexity of Algorithm 1 is $O(TN^2d)$, where $T$ denotes the number of outer for-loop iterations. For further discussion regarding the cost of the optimization (see: Appendix: Sec 9.8). Detailed code implementation is available at `https://github.com/HL-hanlin/OMC`.

---

**Algorithm 1:** Near Orthogonal Monte Carlo: opt-NOMC variant

**Input**: Parameter $\delta, \eta, T$ ;

**Output**: randomly rotated ensemble $\omega_{\mathbf{i}}^{(T)}$ for $i = 1, 2, ..., N$ ;

Initialize $\omega_{\mathbf{i}}^{(0)} (i = 1, 2, ..., N)$ with multiple orthogonal blocks in B-OMC

**for** $t = 0, 1, 2, ..., T-1$ **do**

> Calculate Energy Function $E(\omega_i^{(t)}, \omega_j^{(t)}) = \frac{\delta}{\delta + \|\omega_i^{(t)} - \omega_j^{(t)}\|_2^2}$ for $i \neq j \in \{1, ..., N\}$ ;
>
> **for** $i = 1, 2, ..., N$ **do**
>
> > Compute gradients $F_i^{(t)} = \frac{\partial \sum_{i \neq j} E(\omega_{\mathbf{i}}^{(t)}, \omega_{\mathbf{j}}^{(t)})}{\partial \omega_{\mathbf{i}}^{(t)}}$ ;
> >
> > Update $\omega_{\mathbf{i}}^{(t+1)} \leftarrow \omega_{\mathbf{i}}^{(t)} - \eta F_i^{(t)}$;
> >
> > Normalize $\omega_{\mathbf{i}}^{(t+1)} \leftarrow \frac{\omega_{\mathbf{i}}^{(t+1)}}{\|\omega_{\mathbf{i}}^{(t+1)}\|_2}$ ;
>
> **end**

**end**

---

## 4.2 Algorithm alg-NOMC

As above, without loss of generality we will assume here that $\mathcal{D} = \text{Unif}(\mathcal{S}^{d-1})$ since, as we mentioned above, we can obtain samples for general isotropic $\mathcal{D}$ from the one for $\text{Unif}(\mathcal{S}^{d-1})$ by simple length renormalization. Note that in that setting we can quantify how well the samples from the ensemble $\Omega$ are spread by computing $\mathcal{A}(\Omega) \stackrel{\text{def}}{=} \max_i |\omega_i^\top \omega_j|$. It is a standard fact from probability theory that for base MC samples $\mathcal{A}(\Omega) = \Theta(r^{\frac{1}{2}} d^{-\frac{1}{2}} \sqrt{\log(d)})$ with high probability if the size of $\Omega$ satisfies: $|\Omega| = d^r$ and that is the case also for B-OMC. The question arises: can we do better ?

It turns out that the answer is provided by the theory of algebraic varieties over finite fields. Without loss of generality, we will assume that $d = 2p$, where $p$ is prime. We will encode samples from our structured ensembles via complex-valued functions $g_{c_1,...,c_r} : \mathbb{F}_p \to \mathbb{C}$, given as

$$g_{c_1,...,c_r}(x) = \frac{1}{\sqrt{p}} \exp(\frac{2\pi i(c_r x^r + ... + c_1 x)}{p}), \tag{8}$$

where $\mathbb{F}_p$ and $\mathbb{C}$ stand for the field of residues modulo $p$ and a field of complex numbers respectively and $c_1, ..., c_r \in \mathbb{F}_p$. The encoding $\mathbb{C}^{\mathbb{F}_p} \to \mathbb{R}^d$ is as follows:

$$g_{c_1,...,c_r}(x) \to \mathbf{v}(c_1, ..., c_r) \stackrel{\text{def}}{=} (a_1, b_1, ..., a_p, b_p)^\top \in \mathbb{R}^d, \tag{9}$$

where: $g_{c_1,...,c_r}(j-1) = a_j + ib_j$. Using Weil conjecture for curves, one can show [41] that:

**Lemma 3** (NOMC via algebraic varieties). *If* $\Omega = \{\mathbf{v}(c_1, ..., c_r)\}_{c_1,...,c_r \in \mathbb{F}(p)} \in S^{d-1}$, *then* $|\Omega| = p^r$, *and furthermore* $\mathcal{A}(\Omega) \leq (r-1)p^{-\frac{1}{2}}$.

Thus we see that we managed to get rid of the $\sqrt{\log(d)}$ factor as compared to base MC samples and consequently, obtain better quality ensemble. As for opt-NOMC, before returning samples, we apply random rotation to the entire ensemble. But in contrary to opt-NOMC, in this construction we avoid any optimization, and any more expensive (even one time) computations.

## 5    Experiments

We empirically tested NOMCs in two settings: **(1)** kernel approximation via random feature maps and **(2)** estimating sliced Wasserstein distances, routinely used in generative modeling [42]. For **(1)**, we tested the effectiveness of NOMCs for RBF kernels, non-RBF shift-invariant kernels as well as several PNG kernels. For **(2)**, we considered different classes of multivariate distributions. As we have explained in Sec. 2, the sliced Wasserstein distance for two distributions $\eta, \mu$ is given as:

$$\text{SWD}(\eta, \mu) = (\mathbb{E}_{\mathbf{u} \sim \text{Unif}(\mathcal{S}^{d-1})}[\text{WD}_p^p(\eta_{\mathbf{u}}, \mu_{\mathbf{u}})])^{\frac{1}{p}}. \tag{10}$$

In our experiment we took $p = 2$. We compared against three other methods: **(a)** base Monte Carlo (MC), **(b)** Quasi Monte Carlo applying Halton sequences (QMC) [7] and block orthogonal MC (B-OMC). Additional experimental details are in the Appendix (Sec. 9.10). The results are presented in Fig. 2 and Fig. 3. Empirical MSEs were computed by averaging over $k = 450$ independent experiments. Our NOMC method clearly outperforms other algorithms. For kernel approximation NOMC provides the best accuracy for 7 out of 8 different classes of kernels and for the remaining one it is the second best. For SWD approximation, NOMC provides the best accuracy **for all** 8 **classes** of tested distributions. To the best of our knowledge, NOMC is the first method outperforming B-OMC.

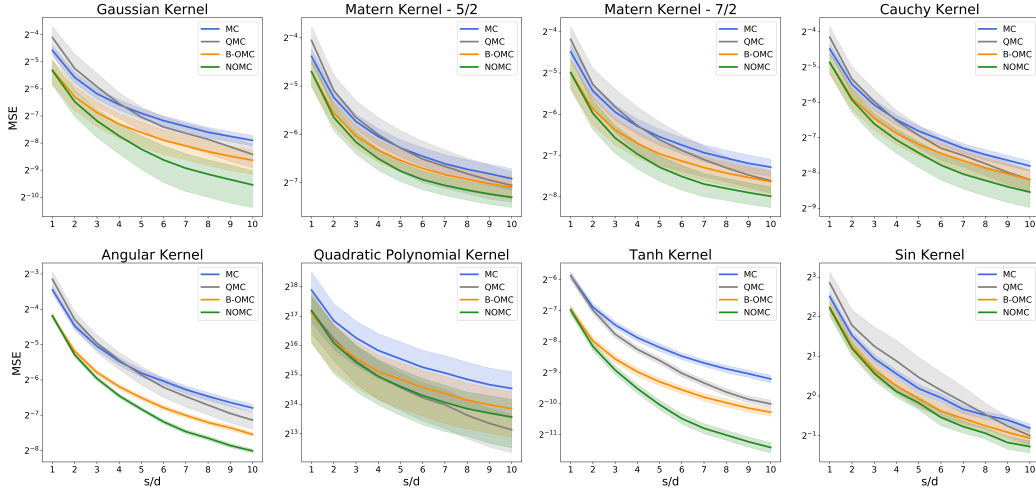

Figure 2: Comparison of MSEs of estimators using different sampling methods: MC, QMC, B-OMC and our NOMC. First four tested kernels are shift-invariant (first three are even RBFs) and last four are PNGs with name indicating nonlinear mapping $h$ used (see: Sec. 2). On the x-axis: the number of blocks (i.e. the ratio of the number of samples $D$ used and data dimensionality $d$). Shaded region corresponds to $0.5 \times \text{std}$.

## 6    Conclusion

In this paper we presented first general theory for the prominent class of orthogonal Monte Carlo (OMC) estimators (used on a regular basis for variance reduction), by discovering an intriguing connection with the theory of negatively dependent random variables. In particular, we give first results for general nonlinear mappings and for **all** RBF kernels as well as first uniform convergence guarantees for OMCs. Inspired by developed theory, we also propose new Monte Carlo algorithm based on near-orthogonal samples (NOMC) that outperforms previous SOTA in the notorious setting, where number of required samples exceeds data dimensionality.

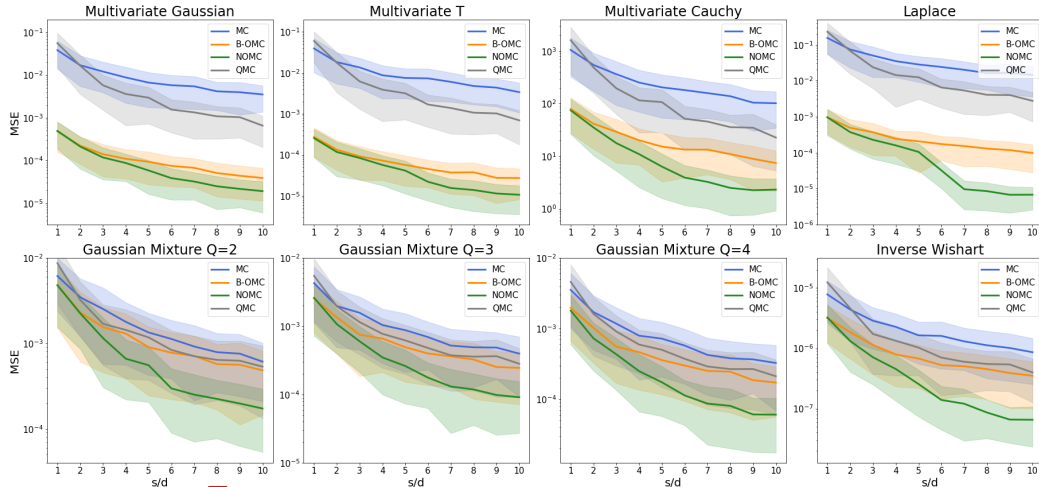

Figure 3: As in Fig. 2, but this time we compare estimators of sliced Wasserstein distances (SWDs) between two distributions taken from a class which name is given above the plot.

# 7 Broader Impact

**General Nonlinear Models:** Understanding the impact of structured Monte Carlo methods leveraging entangled ensembles for general nonlinear models is of crucial importance in machine learning and should guide the research on the developments of new more sample-efficient and accurate MC methods. We think about our results as a first step towards this goal.

**Uniform Convergence Results:** Our uniform convergence results for OMCs from Sec. 3.1.1 are the first such guarantees for OMC methods that can be applied to obtain strong downstream guarantees for OMCs. We demonstrated it on the example of kernel ridge regression, but similar results can be derived for other downstream applications such as kernel-SVM. They are important since in particular they provide detailed guidance on how to choose in practice the number of random features (see: the asymptotic formula for the number of samples in Theorem 3).

**Evolutionary Strategies with Structured MC:** We showed the value of our NOMC algorithm in Sec 5 for kernel and SWD approximation, but the method can be applied as a general tool in several downstream applications, where MC sampling from isotropic distributions is required, in particular in evolutionary strategies (ES) for training reinforcement learning policies [16]. ES techniques became recently increasingly popular as providing state-of-the-art algorithms for tasks of critical importance in robotics such as end-to-end training of high-frequency controllers [22] as well as training adaptable meta-policies [39]. ES methods heavily rely on Monte Carlo estimators of gradients of Gaussians smoothings of certain classes of functions. This makes them potential beneficiaries of new developments in the theory of Monte Carlo sampling and consequently, new Monte Carlo algorithms such as NOMC.

**Algebraic Monte Carlo:** We also think that proposed by us NOMC algorithm in its algebraic variant is one of a very few effective ways of incorporating deep algebraic results into the practice of MC in machine learning. Several QMC methods rely on number theory constructions, but, as we presented, these are much less accurate and in practice not competitive with other structured methods. Not only does our alg-NOMC provide strong theoretical foundations, but it gives additional substantial accuracy gains on the top of already well-optimized methods with no additional computational cost.

# 8 Acknowledgements

We would like to thank Zirui Xu and Ayoujil Jad for their collaboration on putting forward the idea of opt-NOMC, and thank Noemie Perivier, Yan Chen, Haofeng Zhang and Xinyu Zhang for their exploration of orthogonal features for downstream tasks.

# 9 Funding Disclosure

The authors Han Lin, Haoxian Chen, Tianyi Zhang, Clement Laroche are Columbia University students, and Krzysztof Choromanski works in Google and Columbia University. The work in this paper is not funded or supported by any third party.

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
