[Supplementary Material]

# APPENDIX A: Demystifying Orthogonal Monte Carlo and Beyond - Proofs of Theoretical Results

For the convenience of Reader, here we restate the theorems first and then present their proofs.

## 8.1 Proof of Lemma 1

*Proof.* From the definition of negative dependence, what we need to prove is:

$$\mathbb{P}[\bigcap_i^d (|w_i^{\text{ort}\top} \mathbf{z}| \leq \tilde{x}_i)] \leq \prod_i^d \mathbb{P}[|w_i^{\text{ort}\top} \mathbf{z}| \leq \tilde{x}_i] \tag{11}$$

$$\mathbb{P}[\bigcap_i^d (|w_i^{\text{ort}\top} \mathbf{z}| \geq \tilde{x}_i)] \leq \prod_i^d \mathbb{P}[|w_i^{\text{ort}\top} \mathbf{z}| \geq \tilde{x}_i] \tag{12}$$

where we use $\tilde{x}_i$ to represent a different value than the original $x_i$, which should be $f^{-1}(x_i)$. We will illustrate how to prove the first inequality here since the other can be proved accordingly.

Firstly, we know from the definition of isotropic probabilistic distribution that if $w_i^{\text{ort}} \sim \mathcal{D}$, then $w_i^{\text{ort}}$ can be rewritten as $w_i^{\text{ort}} = v_i^{\text{ort}} l_i$, where $v_i^{\text{ort}}$ has unit length, and $l_i$ is taken independently from $v_i^{\text{ort}}$, which represents the length scalar [15]. So we need to prove the following:

$$\mathbb{P}[\bigcap_i^d (|v_i^{\text{ort}\top} \mathbf{z}| \leq \frac{\tilde{x}_i}{l_i})] \leq \prod_i^d \mathbb{P}[|v_i^{\text{ort}\top} \mathbf{z}| \leq \frac{\tilde{x}_i}{l_i}] \tag{13}$$

But actually since negatively dependence should holds for any $x_i \in \mathbb{R}$, so it actually does not matters which scalar we use in the right hand side of each part of the probability inequality. So we will continue to use $x_i$ instead of $\frac{\tilde{x}_i}{l_i}$ in the following proof.

Furthermore, we assume $\|\mathbf{z}\|_2 = 1$ without loss of generality. Proof for the cases when $x_i \geq 1$ or $x_i \leq 0$ is trivial under such assumption, so we will only concentrate on the case when $0 < x_j < 1$. Here, we can use a second trick for distribution transformation. We regard $v_1^{\text{ort}}, v_2^{\text{ort}}, ..., v_d^{\text{ort}}$ as fixed, and $\mathbf{z}$ as a random rotation vector, so that we can replace $v_1^{\text{ort}}, v_2^{\text{ort}}, ..., v_d^{\text{ort}}$ as $e_1, e_2, ..., e_d$ and $\mathbf{z}$ be a unit length vector uniformly distributed on the $\mathcal{S}^{d-1}$. After such transformation, the distribution of $|v_i^{\text{ort}\top} \mathbf{z}|$ will be equivalent to $\frac{|e_i^\top g|}{\|g\|_2} = \frac{g_i}{\|g\|_2}$, where $g$ is a gaussian vector, and $g_i$ is its length of projection onto the $i^{\text{th}}$ coordinate.

So the problem we need to prove is transformed to the following inequality:

$$\mathbb{P}[\bigcap_i^d (\frac{g_i}{\|g\|_2} \leq x_i)] \leq \prod_i^d \mathbb{P}[\frac{g_i}{\|g\|_2} \leq x_i] \tag{14}$$

From the rule of conditional probability, the LHS can be transformed to:

$$\mathbb{P}[\frac{g_1}{\|g\|_2} \leq x_1] \mathbb{P}[\frac{g_2}{\|g\|_2} \leq x_2 | \frac{g_1}{\|g\|_2} \leq x_1] \mathbb{P}[\frac{g_3}{\|g\|_2} \leq x_3 | (\frac{g_1}{\|g\|_2} \leq x_1) \cap (\frac{g_2}{\|g\|_2} \leq x_2)]... \tag{15}$$

until the conditional probability of $\frac{g_d}{\|g\|_2}$ on all $\frac{g_i}{\|g\|_2}$ for $i = 1, 2, ..., d-1$.

Therefore, we conclude that we only need to prove the following for each corresponding term $i$:

$$\mathbb{P}[\frac{g_i}{\|g\|_2} \leq x_i | \bigcap_{j=1}^{i-1} (\frac{g_j}{\|g\|_2} \leq x_j)] \leq P(\frac{g_i}{\|g\|_2} \leq x_i) \tag{16}$$

We note that $\frac{g_j}{\|g\|_2} \leq x_j$ is equivalent to $\frac{g_j^2}{\|g\|_2^2} \leq x_j^2$. So for each $j < i$ we have:

$$g_j^2 \leq x_j^2 g_i^2 + x_j^2 (g_1^2 + ... + g_{i-1}^2 + g_{i+1}^2 + ... + g_d^2) \tag{17}$$

which can be rewritten as:

$$g_i{}^2 \geq \frac{g_j{}^2}{x_j{}^2} - (g_1{}^2 + ... + g_{i-1}{}^2 + g_{i+1}{}^2 + ... + g_d{}^2) \tag{18}$$

We can also rewrite from $\frac{g_i}{\|g\|_2} \leq x_i$ and derive:

$$g_i{}^2 \leq \frac{x_i{}^2}{(1 - x_i{}^2)}(g_1{}^2 + ... + g_{i-1}{}^2 + g_{i+1}{}^2 + ... + g_d{}^2) \tag{19}$$

Therefore,

$$\mathbb{P}[\frac{g_i}{\|g\|_2} \leq x_i | \bigcap_{j=1}^{i-1}(\frac{g_j}{\|g\|_2} \leq x_j)] = \mathbb{P}[g_i{}^2 \leq \frac{x_i{}^2}{(1 - x_i{}^2)}(g_1{}^2 + ... + g_{i-1}{}^2 + g_{i+1}{}^2 + ... + g_d{}^2)$$

$$| \bigcap_{j=1}^{i-1}(g_i{}^2 \geq \frac{g_j{}^2}{x_j{}^2} - (g_1{}^2 + ... + g_{i-1}{}^2 + g_{i+1}{}^2 + ... + g_d{}^2))]$$

$$\leq \mathbb{P}[g_i{}^2 \leq \frac{x_i{}^2}{(1 - x_i{}^2)}(g_1{}^2 + ... + g_{i-1}{}^2 + g_{i+1}{}^2 + ... + g_d{}^2)] = \mathbb{P}[\frac{g_i}{\|g\|_2} \leq x_i]$$
$$\tag{20}$$

which finishes our proof of negative dependence.

$\square$

We can conclude that:

**Lemma 1** (ND for OMC-samples and monotone functions). *For an isotropic distribution $\mathcal{D}$ on $\mathbb{R}^d$ and orthogonal ensemble: $\omega_1^{\mathrm{ort}}, ..., \omega_d^{\mathrm{ort}}$ with $\omega_i^{\mathrm{ort}} \sim \mathcal{D}$, random variables: $X_1, ..., X_d$ defined as: $X_i = |\omega_i^{\mathrm{ort}} \mathbf{z}|$ are negatively dependent for any fixed $\mathbf{z} \in \mathbb{R}^d$.*

## 8.2 Proof of Lemma 2

To prove Lemma 2, we will use the following result [4], [25]:

**Lemma 4.** *Let $X_1, \ldots, X_n$ be negatively dependent random variables, then:*

- *If $f_1, \ldots, f_n$ is a sequence of measurable functions which are all monotone non-decreasing (or all are monotone non-increasing), then $f_1(X_1), \ldots, f_n(X_n)$ are also negatively dependent random variables.*

- $\mathbb{E}[X_1 \ldots X_n] \leq \mathbb{E}[X_1] \ldots \mathbb{E}[X_n]$, *provided the expectation exist.*

**Lemma 2.** *Assume that $f$ is a function from the class **F1**. Let $X_i = f(\omega_i^{\mathrm{ort}}{}^\top \mathbf{z})$ for $i = 1, ..., n$, and let $\lambda$ be a non-positive (or non-negative) real number. Then the following holds:*

$$\mathbb{E}[\mathrm{e}^{\lambda \sum_{i=1}^m X_i}] \leq \prod_{i=1}^{m} \mathbb{E}[\mathrm{e}^{\lambda X_i}].$$

*Proof.* By Lemma 1 and the first point iof Lemma 4, we know that $X_1, ..., X_n$ are negatively dependent. Then, by applying the second point in Lemma 4, we know that:

$$\mathbb{E}[f_1(X_1) \ldots f_n(X_n)] \leq \mathbb{E}[f_1(X_1)] \ldots \mathbb{E}[f_n(X_n)] \tag{21}$$

If $\lambda \geq 0$, we can take a non-decreasing function $f_i(X_i) = e^{\lambda X_i}$ for each $i$, then:

$$\mathbb{E}[\exp(\lambda \sum_{i=1}^{m} X_i)] \leq \prod_{i=1}^{m} \mathbb{E}[e^{\lambda X_i}] \tag{22}$$

Similarly, if $\lambda \leq 0$, then we can take a non-increasing function $f_i(X_i) = e^{\lambda X_i}$, and this inequality will also be true. Actually, we say that $X_1, ... X_n$ are acceptable if the inequality $\mathbb{E}[\exp(\lambda \sum_{i=1}^{m} X_i)] \leq \prod_{i=1}^{m} \mathbb{E}[e^{\lambda X_i}]$ holds for any real $\lambda$ [4]. $\square$

## 8.3 Proof of Corollary 1

**Corollary 1** (exponentials of OMCs and MCs). *Let $\mathbf{z} \in \mathbb{R}^d$ and assume that function $f : \mathbb{R} \to \mathbb{R}$ is from the class **F1**. Take an isotropic distribution $\mathcal{D}$ on $\mathbb{R}^d$, an ensemble of independent samples $\omega_1^{\mathrm{iid}}, ..., \omega_s^{\mathrm{iid}}$ and an orthogonal ensemble $\omega_1^{\mathrm{ort}}, ..., \omega_s^{\mathrm{ort}}$ giving rise to base MC estimator $\widehat{F}_{f,\mathcal{D}}^{\mathrm{iid}}(\mathbf{z})$ of $\mathbb{E}_{\omega \sim \mathcal{D}}[f(\omega^\top \mathbf{z})]$ and to its orthogonal version $\widehat{F}_{f,\mathcal{D}}^{\mathrm{ort}}(\mathbf{z})$. Then the following is true for any $\lambda$:*

$$\mathbb{E}[e^{\lambda \widehat{F}_{f,\mathcal{D}}^{\mathrm{ort}}(\mathbf{z})}] \leq \mathbb{E}[e^{\lambda \widehat{F}_{f,\mathcal{D}}^{\mathrm{iid}}(\mathbf{z})}]. \tag{4}$$

*Proof.* From Lemma 1 and Lemma 2, we can derive directly that if the function $f$ is monotone increasing (or decreasing) in $|\omega_i^\top \mathbf{z}|$, and we define $\widehat{F}_{f,\mathcal{D}}^{\mathrm{ort}}(\mathbf{z})$ and $\widehat{F}_{f,\mathcal{D}}^{\mathrm{iid}}(\mathbf{z})$ as the orthogonal and iid estimates for $\mathbb{E}_{\omega \sim \mathcal{D}}[f_{\mathcal{Z}}(\omega)]$, then the ND of $|\omega_1^{ort\top} \mathbf{z}|, ..., |\omega_d^{ort\top} \mathbf{z}|$ implies $\forall \lambda \in \mathbf{R}$ and $s = d$:

$$\mathbb{E}[\exp(\lambda \widehat{F}_{f,\mathcal{D}}^{\mathrm{ort}}(\mathbf{z}))] \leq \prod_{i=1}^d \mathbb{E}[e^{\lambda \widehat{F}_{f,\mathcal{D}}^{\mathrm{ort}}(\mathbf{z})}] = \prod_{i=1}^d \mathbb{E}[e^{\lambda \widehat{F}_{f,\mathcal{D}}^{\mathrm{iid}}(\mathbf{z})}] \tag{23}$$

which is exactly the inequality in this Corollary.

For $s = kd$ where $k$ is a multiplier larger than 1, we can define $\widehat{F}_{f,\mathcal{D}}^{\mathrm{ort}}(\mathbf{z})$ as the estimator constructed by stacking $k$ independent orthogonal blocks together with dimension $d$, and $\widehat{F}_{f,\mathcal{D}}^{\mathrm{iid}}(\mathbf{z})$ as the base estimator with $s$ samples. The proof in such case is trivial since we can decompose $\mathbb{E}[\exp(\lambda \widehat{F}_{f,\mathcal{D}}^{\mathrm{ort}}(\mathbf{z}))]$ into the multiplication of $k$ expectations of independent blocks, and then use Eq. 23 again.

$\square$

## 8.4 Proof of Theorem 1

**Theorem 1** (OMC-bounds surpassing MC-bounds for the **F1**-class). *Denote by $\mathrm{MSE}$ a mean squared error of the estimator, by $s$ the number of MC samples used and let $X = f(\omega^\top \mathbf{z})$ for $\omega \sim \mathcal{D}$. Then under assumptions as in Corollary 1, OMC leads to the unbiased estimator satisfying for $\epsilon > 0$:*

$$\mathbb{P}[|\widehat{F}_{f,\mathcal{D}}^{\mathrm{ort}}(\mathbf{z}) - F_{f,\mathcal{D}}(\mathbf{z})| \geq \epsilon] \leq p(\epsilon), \tag{5}$$

*where $p(\epsilon)$ is defined as: $p(\epsilon) \stackrel{\text{def}}{=} \exp(-s\mathcal{L}_X(F_{f,\mathcal{D}}(\mathbf{z})+\epsilon) + \exp(-s\mathcal{L}_X(F_{f,\mathcal{D}}(\mathbf{z})-\epsilon)$ for unbounded $f$, and is defined as $p(\epsilon) \stackrel{\text{def}}{=} 2\exp(-\frac{2s\epsilon^2}{(b-a)^2})$ for bounded $f \in [a, b]$, which is a standard upper bound on $\mathbb{P}[|\widehat{F}_{f,\mathcal{D}}^{\mathrm{iid}}(\mathbf{z}) - F_{f,\mathcal{D}}(\mathbf{z})| \geq \epsilon]$. Furthermore: $\mathrm{MSE}(\widehat{F}_{f,\mathcal{D}}^{\mathrm{ort}}(\mathbf{z})) \leq \mathrm{MSE}(\widehat{F}_{f,\mathcal{D}}^{\mathrm{iid}}(\mathbf{z}))$.*

*Proof.* Let's first work on the case when function $f$ is bounded. In such case, we can apply Chernoff-Hoeffdings inequality for iid estimators to $p(\epsilon)$, which is $2\exp(-\frac{2s\epsilon^2}{(b-a)^2})$.

For $\lambda > 0, \epsilon \in \mathbb{R}$, we apply Markov inequality here:

$$\mathbb{P}[\widehat{F}_{f,\mathcal{D}}^{\mathrm{ort}}(\mathbf{z}) - F_{f,\mathcal{D}}(\mathbf{z}) \geq \epsilon] = \mathbb{P}[e^{\lambda(\widehat{F}_{f,\mathcal{D}}^{\mathrm{ort}}(\mathbf{z}) - F_{f,\mathcal{D}}(\mathbf{z}))} \geq e^{\lambda\epsilon}]$$
$$\leq e^{-\lambda\epsilon}\mathbb{E}[e^{\lambda(\widehat{F}_{f,\mathcal{D}}^{\mathrm{ort}}(\mathbf{z}) - F_{f,\mathcal{D}}(\mathbf{z}))}] = e^{-\lambda\epsilon}e^{-\lambda F_{f,\mathcal{D}}(\mathbf{z})}\mathbb{E}[e^{\lambda \widehat{F}_{f,\mathcal{D}}^{\mathrm{ort}}(\mathbf{z})}] \tag{24}$$

Similarly, for iid estimator, we have:

$$\mathbb{P}[\widehat{F}_{f,\mathcal{D}}^{\mathrm{iid}}(\mathbf{z}) - F_{f,\mathcal{D}}(\mathbf{z}) \geq \epsilon] = e^{-\lambda\epsilon}e^{-\lambda F_{f,\mathcal{D}}(\mathbf{z})}\mathbb{E}[e^{\lambda \widehat{F}_{f,\mathcal{D}}^{\mathrm{iid}}(\mathbf{z})}] \tag{25}$$

From Corollary 1, we know directly that orthogonal estimator has better upper bound than iid estimator.

For $\lambda < 0$,

$$\mathbb{P}[\widehat{F}_{f,\mathcal{D}}^{\mathrm{ort}}(\mathbf{z}) - F_{f,\mathcal{D}}(\mathbf{z}) \leq -\epsilon] = \mathbb{P}[e^{\lambda(\widehat{F}_{f,\mathcal{D}}^{\mathrm{ort}}(\mathbf{z}) - F_{f,\mathcal{D}}(\mathbf{z}))} \geq e^{\lambda\epsilon}] \tag{26}$$

$\mathbb{E}[e^{\lambda \widehat{F}_{f,\mathcal{D}}^{\mathrm{ort}}(\mathbf{z})}] \leq \mathbb{E}[e^{\lambda \widehat{F}_{f,\mathcal{D}}^{\mathrm{iid}}(\mathbf{z})}]$ in Corollary 1 also guarantees better lower bound than iid estimator.

By combining these two cases, we know that $\mathbb{P}[|\widehat{F}^{\mathrm{ort}}_{f,\mathcal{D}}(\mathbf{z}) - F_{f,\mathcal{D}}(\mathbf{z})| \geq \epsilon]$ has better bound than $\mathbb{P}[|\widehat{F}^{\mathrm{iid}}_{f,\mathcal{D}}(\mathbf{z}) - F_{f,\mathcal{D}}(\mathbf{z})| \geq \epsilon]$.

Then for unbounded function $f$, we can apply Cramer-Chernoff bound to $p(\epsilon)$. We rewrite several steps here to show:

$$p(\epsilon) = \exp\{-s(\mathcal{L}_X(F_{f,\mathcal{D}}(\mathbf{z})) + \epsilon)\} + \exp\{-s(\mathcal{L}_X(F_{f,\mathcal{D}}(\mathbf{z})) - \epsilon)\}$$

which is the bound for iid estimator.

$$\mathbb{P}[|\widehat{F}^{\mathrm{iid}}_{f,\mathcal{D}}(\mathbf{z}) - F_{f,\mathcal{D}}(\mathbf{z})| \geq \epsilon]$$

$$= \mathbb{P}[\widehat{F}^{\mathrm{iid}}_{f,\mathcal{D}}(\mathbf{z}) - F_{f,\mathcal{D}}(\mathbf{z}) \geq \epsilon] + \mathbb{P}[\widehat{F}^{\mathrm{iid}}_{f,\mathcal{D}}(\mathbf{z}) - F_{f,\mathcal{D}}(\mathbf{z}) \leq -\epsilon]$$

$$= \mathbb{P}[\sum_{i=1}^{s} f_{\mathcal{Z}}(\omega_i^{\mathrm{iid}}) - s F_{f,\mathcal{D}}(\mathbf{z}) \geq s\epsilon] + \mathbb{P}[\sum_{i=1}^{s} f_{\mathcal{Z}}(\omega_i^{\mathrm{iid}}) - s F_{f,\mathcal{D}}(\mathbf{z}) \leq -s\epsilon]$$

$$\leq \exp\{-\sup_{\theta>0}(\theta s\epsilon - \log^{\mathbb{E}[e^{\theta(\sum_{i=1}^{s} f_{\mathcal{Z}}(\omega_i^{\mathrm{iid}}) - s F_{f,\mathcal{D}}(\mathbf{z}))}]})\}$$

$$+ \exp\{-\sup_{\theta<0}(-\theta s\epsilon - \log^{\mathbb{E}[e^{\theta(\sum_{i=1}^{s} f_{\mathcal{Z}}(\omega_i^{\mathrm{iid}}) - s F_{f,\mathcal{D}}(\mathbf{z}))}]})\}$$

$$= \exp\{-s\sup_{\theta>0}(\theta\epsilon - \frac{1}{s}\log^{\mathbb{E}[e^{\theta(\sum_{i=1}^{s} f_{\mathcal{Z}}(\omega_i^{\mathrm{iid}}))}]} + \theta F_{f,\mathcal{D}}(\mathbf{z}))\}$$

$$+ \exp\{-s\sup_{\theta<0}(-\theta\epsilon - \frac{1}{s}\log^{\mathbb{E}[e^{\theta(\sum_{i=1}^{s} f_{\mathcal{Z}}(\omega_i^{\mathrm{iid}}))}]} + \theta F_{f,\mathcal{D}}(\mathbf{z}))\}$$

$$= \exp\{-s\sup_{\theta>0}(\theta\epsilon - \log^{\mathbb{E}[e^{\theta(f_{\mathcal{Z}}(\omega_i^{\mathrm{iid}}))}]} + \theta F_{f,\mathcal{D}}(\mathbf{z}))\}$$

$$+ \exp\{-s\sup_{\theta<0}(-\theta\epsilon - \log^{\mathbb{E}[e^{\theta(f_{\mathcal{Z}}(\omega_i^{\mathrm{iid}}))}]} + \theta F_{f,\mathcal{D}}(\mathbf{z}))\}$$

$$= \exp\{-s\sup_{\theta>0}(\theta(F_{f,\mathcal{D}}(\mathbf{z}) + \epsilon) - \log^{\mathbb{E}[e^{\theta(f_{\mathcal{Z}}(\omega_i^{\mathrm{iid}}))}]})\}$$

$$+ \exp\{-s\sup_{\theta<0}(-\theta(F_{f,\mathcal{D}}(\mathbf{z}) - \epsilon) - \log^{\mathbb{E}[e^{\theta(f_{\mathcal{Z}}(\omega_i^{\mathrm{iid}}))}]})\}$$

$$= \exp\{-s(\mathcal{L}_X(F_{f,\mathcal{D}}(\mathbf{z})) + \epsilon)\} + \exp\{-s(\mathcal{L}_X(F_{f,\mathcal{D}}(\mathbf{z})) - \epsilon)\} \tag{27}$$

where $\mathcal{L}_X(a) = \sup_{\theta>0} \log(\frac{e^{\theta a}}{M_X(\theta)})$ if $a > \mathbb{E}[X]$ and $\mathcal{L}_X(a) = \sup_{\theta<0} \log(\frac{e^{\theta a}}{M_X(\theta)})$ if $a < \mathbb{E}[X]$.

The proof for the superiority of orthogonal estimator is similar as above, and we include it here for completeness.

For $\lambda > 0, \epsilon \in \mathbb{R}$, we have:

$$\mathbb{P}[\widehat{F}^{\mathrm{ort}}_{f,\mathcal{D}}(\mathbf{z}) - F_{f,\mathcal{D}}(\mathbf{z}) \geq \epsilon] \leq \exp\{-\sup_{\theta>0}(\lambda\epsilon - \log^{\mathbb{E}[e^{\lambda(\widehat{F}^{\mathrm{ort}}_{f,\mathcal{D}}(\mathbf{z}) - F_{f,\mathcal{D}}(\mathbf{z}))}]})\}$$

$$= \exp\{-\sup_{\theta>0}(\lambda(\epsilon + F_{f,\mathcal{D}}(\mathbf{z})) - \log^{\mathbb{E}[e^{\lambda(\widehat{F}^{\mathrm{ort}}_{f,\mathcal{D}}(\mathbf{z}))}]})\} \tag{28}$$

We can derive similarly such probability bound for iid estimator. Then by applying Corollary 1, we know that $\log^{\mathbb{E}[e^{\lambda\widehat{F}^{\mathrm{ort}}_{f,\mathcal{D}}(\mathbf{z})}]} \leq \log^{\mathbb{E}[e^{\lambda\widehat{F}^{\mathrm{iid}}_{f,\mathcal{D}}(\mathbf{z})}]}$. With such relationship, we know directly that orthogonal estimator has better upper bound than iid estimator.

The same follows for $\lambda < 0$. And we can combine these two cases and derive that $\mathbb{P}[|\widehat{F}^{\mathrm{ort}}_{f,\mathcal{D}}(\mathbf{z}) - F_{f,\mathcal{D}}(\mathbf{z})| \geq \epsilon]$ has better bound than $\mathbb{P}[|\widehat{F}^{\mathrm{iid}}_{f,\mathcal{D}}(\mathbf{z}) - F_{f,\mathcal{D}}(\mathbf{z})| \geq \epsilon]$.

Finally, for the MSE of the iid estimator, we know from the independence of $(\omega_i)_{i=1}^{s}$ that:

$$\mathrm{MSE}(\widehat{F}^{\mathrm{iid}}_{f,\mathcal{D}}(\mathbf{z})) = \frac{1}{s^2} \sum_{i=1}^{s} \mathrm{Var}[f(\omega_i^\top \mathbf{z})] = \frac{1}{s} \mathrm{Var}[f(\omega_1^\top \mathbf{z})] \tag{29}$$

We can also decompose the MSE of orthogonal estimator as:

$$\text{MSE}(\widehat{F}_{f,\mathcal{D}}^{\text{ort}}(\mathbf{z})) = \frac{1}{s}\text{Var}[f(\omega_1^\top \mathbf{z})] + \frac{1}{s^2}\sum_{i\neq j}(\mathbb{E}[f(\omega_i^\top \mathbf{z})f(\omega_j^\top \mathbf{z})] - \mathbb{E}[f(\omega_i^\top \mathbf{z})]\mathbb{E}[f(\omega_j^\top \mathbf{z})]) \quad (30)$$

Since a subset of two ND variables are also ND, the ND of $(f(\omega_i^\top \mathbf{z}))_{i=1}^s$ implies that the second part of $\text{MSE}(\widehat{F}_{f,\mathcal{D}}^{\text{ort}}(\mathbf{z}))$ is negative, which completes the proof.

$\square$

We further notice that since $\mathcal{D}$ is an isotropic probability distribution on $\mathbb{R}^d$ which is rotation invariant, then for $\omega \sim \mathcal{D}$ and an odd function $\text{odd}[f]$, we have $\mathbb{P}(\omega) = \mathbb{P}(-\omega)$ and $\text{odd}[f](\omega^\top \mathbf{z}) = -\text{odd}[f](-\omega^\top \mathbf{z})$. Therefore,

$$F_{\text{odd}[f],\mathcal{D}} = \mathbb{E}_{\omega\sim\mathcal{D}}[\text{odd}[f](\omega^\top \mathbf{z})] = \int_{\mathcal{D}}\text{odd}[f](\omega^\top \mathbf{z})d\mathbb{P}(\omega) = 0 \quad (31)$$

## 8.5 Proof of Theorem 2

**Theorem 2** (Exponential bounds for OMCs and **F2**/**F3** classes). *Let $\mathbf{z} \in \mathbb{R}^d$ and assume that function $f : \mathbb{R} \to \mathbb{R}$ is from the class **F2** or **F3**. Then for $\epsilon > 0$:*

$$\mathbb{P}[|\widehat{F}_{f,\mathcal{D}}^{\text{ort}}(\mathbf{z}) - F_{f,\mathcal{D}}(\mathbf{z})| \geq \epsilon] \leq p(\epsilon) \stackrel{\text{def}}{=} u^+ + u^-, \quad (6)$$

*where $u^{+/-} \stackrel{\text{def}}{=} \exp(-s\mathcal{L}_{X^{+/-}}(F_{f,\mathcal{D}}(\mathbf{z}) + \frac{\epsilon}{2})) + \exp(-s\mathcal{L}_{X^{+/-}}(F_{f,\mathcal{D}}(\mathbf{z}) - \frac{\epsilon}{2}))$, and $X^{+/-}$ is defined as: $X^{+/-} \stackrel{\text{def}}{=} f^{+/-}$ if $f$ is from **F2** and as: $X^{+/-} \stackrel{\text{def}}{=} (\text{even}[f])^{+/-}$ if $f$ is from **F3**. As before, in the bounded case we can simplify $u^+$ and $u^-$ to: $u^{+/-} \stackrel{\text{def}}{=} 2\exp(-\frac{s\epsilon^2}{2(b^{+/-}-a^{+/-})^2})$, where $a^+, b^+, a^-, b^-$ are lower and upper bounds such that: $f^+ \in [a^+, b^+]$ and $f^- \in [a^-, b^-]$ if $f$ is from **F2** or $(\text{even}[f])^+ \in [a^+, b^+]$ and $(\text{even}[f])^- \in [a^-, b^-]$ if $f$ is from **F3**. Furthermore, if $(\text{even}[f])^+ = 0$ or $(\text{even}[f])^- = 0$, we can tighten that bound using upper bound from Theorem 1 and thus, establish better concentration bounds than for base MC.*

*Proof.* Firstly, we decompose the estimator into increasing and decreasing parts as stated in **F2**:

$$\widehat{F}_{f,\mathcal{D}}^{\text{ort}}(\mathbf{z}) = \frac{1}{s}\sum_{i=1}^s f(|\omega_i^{\text{ort}\top}\mathbf{z}|) = \frac{1}{s}\sum_{i=1}^s f^+(|\omega_i^{\text{ort}\top}\mathbf{z}|) + \frac{1}{s}\sum_{i=1}^s f^-(|\omega_i^{ort\top}\mathbf{z}|)$$
$$\stackrel{\text{def}}{=} \widehat{F}_{f,\mathcal{D}}^{\text{ort},+}(\mathbf{z}) + \widehat{F}_{f,\mathcal{D}}^{\text{ort},-}(\mathbf{z}) \quad (32)$$

which are ND respectively.

For bounded function $f$, we can apply Chernoff–Hoeffding inequalities for ND random variables [21] and have:

$$\mathbb{P}[|\widehat{F}_{f,\mathcal{D}}^{\text{ort},+}(\mathbf{z}) - F_{f,\mathcal{D}}^+(\mathbf{z})| \geq \epsilon] \leq 2\exp(-\frac{2s\epsilon^2}{(b^+ - a^+)^2}) \quad (33)$$

$$\mathbb{P}[|\widehat{F}_{f,\mathcal{D}}^{\text{ort},-}(\mathbf{z}) - F_{f,\mathcal{D}}^-(\mathbf{z})| \geq \epsilon] \leq 2\exp(-\frac{2s\epsilon^2}{(b^- - a^-)^2}) \quad (34)$$

Therefore,

$$\mathbb{P}[|\widehat{F}_{f,\mathcal{D}}^{\text{ort}}(\mathbf{z}) - F_{f,\mathcal{D}}(\mathbf{z})| \geq \epsilon]$$
$$= \mathbb{P}[|\widehat{F}_{f,\mathcal{D}}^{\text{ort},+}(\mathbf{z}) + \widehat{F}_{f,\mathcal{D}}^{\text{ort},-}(\mathbf{z}) - F_{f,\mathcal{D}}^+(\mathbf{z}) - F_{f,\mathcal{D}}^-(\mathbf{z})| \geq \epsilon]$$
$$\leq \mathbb{P}[|\widehat{F}_{f,\mathcal{D}}^{\text{ort},+}(\mathbf{z}) - F_{f,\mathcal{D}}^+(\mathbf{z})| + |\widehat{F}_{f,\mathcal{D}}^{\text{ort},-}(\mathbf{z}) - F_{f,\mathcal{D}}^-(\mathbf{z})| \geq \epsilon]$$
$$\leq \mathbb{P}[|\widehat{F}_{f,\mathcal{D}}^{\text{ort},+}(\mathbf{z}) - F_{f,\mathcal{D}}^+(\mathbf{z})| \geq \frac{\epsilon}{2}] + \mathbb{P}[|\widehat{F}_{f,\mathcal{D}}^{\text{ort},-}(\mathbf{z}) - F_{f,\mathcal{D}}^-(\mathbf{z})| \geq \frac{\epsilon}{2}]$$
$$\leq 2\exp(-\frac{s\epsilon^2}{2(b^+ - a^+)^2}) + 2\exp(-\frac{s\epsilon^2}{2(b^- - a^-)^2}) \quad (35)$$

This procedure can be adapted to the case of unbounded function $f$ with similar steps in Theorem 1, so we skip it.

$\square$

## 8.6 Proof of Theorem 3

**Theorem 3** (Uniform convergence for OMCs). *Let $\mathcal{M} \subseteq \mathbb{R}^d$ be compact with diameter $\mathrm{diam}(\mathcal{M})$. Assume that $f$ has Lipschitz constant $L_f$. Then under assumptions as in Theorem 1 / 2, for any $r > 0$:*

$$\mathbb{P}[\sup_{\mathbf{z} \in \mathcal{M}} |\widehat{F}_{f,\mathcal{D}}^{\mathrm{ort}}(\mathbf{z}) - F_{f,\mathcal{D}}(\mathbf{z})| \geq \epsilon] \leq C(\frac{\mathrm{diam}(\mathcal{M})}{r})^d \cdot p(\epsilon/2) + (\frac{2r\sigma L_f}{\epsilon})^2, \qquad (7)$$

*where $\sigma^2 = \mathbb{E}_{\omega \sim D}[\omega^T \omega]$ (i.e. the second moment of $D$), $p$ is as in RHS of inequality from Theorem 1 / 2 and $C > 0$ is a universal constant. In particular, if boundedness conditions from Theorem 1 / 2 are satisfied, one can take: $s = \Theta(\frac{d}{\epsilon^2} \log(\frac{\sigma L_f \mathrm{diam}(\mathcal{M})}{\epsilon}))$ to get uniform $\epsilon$-error approximation with any constant probability (e.g $s = \Theta(\frac{d}{\epsilon^2} \log(\frac{d L_f \mathrm{diam}(\mathcal{M})}{\epsilon}))$ for Gaussian kernel for which $\sigma = d$).*

*Proof.* Motivated by [34], the uniform convergence for OMCs can be proved in the following way. Define $g(\mathbf{z}) = \widehat{F}_{f,\mathcal{D}}^{\mathrm{ort}}(\mathbf{z}) - F_{f,\mathcal{D}}(\mathbf{z})$. Given the definition of $\widehat{F}_{f,\mathcal{D}}^{\mathrm{ort}}(\mathbf{z})$, it is unbiased, i.e. $\mathbb{E}[g(\mathbf{z})] = \mathbb{E}[\widehat{F}_{f,\mathcal{D}}^{\mathrm{ort}}(\mathbf{z}) - F_{f,\mathcal{D}}(\mathbf{z})] = 0$.

Let $\mathcal{M} \subseteq \mathbb{R}^d$ be compact with diameter $\mathrm{diam}(\mathcal{M})$ and $\mathbf{z} \in \mathcal{M}$. We can find a $\epsilon$-net such that it can covers $\mathcal{M}$ with at most $P = (\frac{4\mathrm{diam}(\mathcal{M})}{r})^d$ balls of radius $r$. Denote $\{z_i\}_{i=1}^P$ as the centers of the these balls. If $|g(\mathbf{z}_i)| < \frac{\epsilon}{2}$ and Lipschitz constant $L_g$ of $g$ satisfies: $L_g < \frac{\epsilon}{2r}, \forall i \in [P]$, then $|g(\mathbf{z})| < \epsilon$. By applying the union bound followed by Hoeffding's inequality applied to the anchors in the $\epsilon$-net, we can have the following:

$$\mathbb{P}[\bigcup_{i=1}^P |g(\mathbf{z}_i)| \geq \frac{\epsilon}{2}] \leq P \cdot p(\frac{\epsilon}{2}) \qquad (36)$$

If $f$ is differentiable, $L_g = \max_{\mathbf{z} \in \mathcal{M}} ||\nabla g(\mathbf{z}^*)||$. From the linearity of expectation, we can have $\mathbb{E}[\nabla \widehat{F}_{f,\mathcal{D}}^{\mathrm{ort}}(\mathbf{z})] = \nabla F_{f,\mathcal{D}}(\mathbf{z})$, therefore we can have:

$$\begin{aligned}
\mathbb{E}[L_g^2] &= \mathbb{E}[||\nabla \widehat{F}_{f,\mathcal{D}}^{\mathrm{ort}}(\mathbf{z}^*) - \nabla F_{f,\mathcal{D}}(\mathbf{z}^*)||^2] \\
&= \mathbb{E}[||\nabla \widehat{F}_{f,\mathcal{D}}^{\mathrm{ort}}(\mathbf{z}^*)||^2 + ||\nabla F_{f,\mathcal{D}}(\mathbf{z}^*)||^2 - 2\nabla \widehat{F}_{f,\mathcal{D}}^{\mathrm{ort}}(\mathbf{z}^*)^T \nabla F_{f,\mathcal{D}}(\mathbf{z}^*)] \\
&= \mathbb{E}[||\nabla \widehat{F}_{f,\mathcal{D}}^{\mathrm{ort}}(\mathbf{z}^*)||^2] + \mathbb{E}[||\nabla F_{f,\mathcal{D}}(\mathbf{z}^*)||^2] - 2\mathbb{E}[||\nabla F_{f,\mathcal{D}}(\mathbf{z}^*)||^2] \\
&= \mathbb{E}[||\nabla \widehat{F}_{f,\mathcal{D}}^{\mathrm{ort}}(\mathbf{z}^*)||^2] - \mathbb{E}[||\nabla F_{f,\mathcal{D}}(\mathbf{z}^*)||^2] \\
&= \mathbb{E}[||\nabla \widehat{F}_{f,\mathcal{D}}^{\mathrm{ort}}(\mathbf{z}^*)||^2] - ||\nabla F_{f,\mathcal{D}}(\mathbf{z}^*)||^2 \qquad (37)
\end{aligned}$$

Therefore, $\mathbb{E}[L_g^2] \leq \mathbb{E}[||\nabla \widehat{F}_{f,\mathcal{D}}^{\mathrm{ort}}(z^*)||^2] \leq \mathbb{E}_{\mathcal{D}}[||\omega L_f||^2] = \sigma^2 L_f^2$. Finally, if $f$ is not differentiable, we can obtain exactly the same bound via standard finite-difference analysis.
According to the Markov Inequality, we have the following:

$$\mathbb{P}[L_g \geq \frac{\epsilon}{2r}] \leq (\frac{2r\sigma L_f}{\epsilon})^2. \qquad (38)$$

Thus, by union bound, we can conclude that:

$$\mathbb{P}[\sup_{\mathbf{z} \in \mathcal{M}} |g(\mathbf{z})| \geq \epsilon] \leq (\frac{4\mathrm{diam}(\mathcal{M})}{r})^d \cdot p(\frac{\epsilon}{2}) + (\frac{2r\sigma L_f}{\epsilon})^2, \qquad (39)$$

which is our results for general $f$. Now let us consider the case when $f$ is bounded. For the case of **F1**, we can let $p(\epsilon) = 2\exp(-\frac{2s\epsilon^2}{(b-a)^2})$. Then:

$$\mathbb{P}[\sup_{\mathbf{z} \in \mathcal{M}} |g(\mathbf{z})| \geq \epsilon] \leq 2(\frac{4\mathrm{diam}(\mathcal{M})}{r})^d \exp(-\frac{s\epsilon^2}{2(b-a)^2}) + (\frac{2r\sigma L_f}{\epsilon})^2 \qquad (40)$$

For the case of **F2/F3**, we can have: $p(\epsilon) = 2(\exp(-\frac{s\epsilon^2}{2(b^+-a^+)^2}) + \exp(-\frac{s\epsilon^2}{2(b^--a^-)^2}))$. Then:

$$\mathbb{P}[\sup_{\mathbf{z}\in\mathcal{M}}|g(\mathbf{z})| \geq \epsilon] \leq 2(\frac{4\text{diam}(\mathcal{M})}{r})^d[\exp(-\frac{s\epsilon^2}{8(b^+-a^+)^2}) + \exp(-\frac{s\epsilon^2}{8(b^--a^-)^2})] + (\frac{2r\sigma L_f}{\epsilon})^2$$
(41)

One can take $C = 2 \cdot 4^d$ here. In order to find smallest $s$ such that **F1/F2/F3** can satisfy this bound, we can optimize for $r$ and this is how we get the asymptotic value of the number of samples $s$ that provides $\epsilon$-accuracy (we assume here that bounds on $f$ are constants):

$$s = \Theta(\frac{d}{\epsilon^2}\log(\frac{\sigma L_f \text{diam}(\mathcal{M})}{\epsilon})).$$
(42)

Another case is that $f$ is unbounded, For the case of **F1**, we can let $p(\epsilon) = \exp\{-s(\mathcal{L}_X(F_{f,\mathcal{D}}(\mathbf{z})) + \epsilon)\} + \exp\{-s(\mathcal{L}_X(F_{f,\mathcal{D}}(\mathbf{z})) - \epsilon)\}$. Then:

$$\mathbb{P}[\sup_{\mathbf{z}\in\mathcal{M}}|g(\mathbf{z})| \geq \epsilon] \leq (\frac{4\text{diam}(\mathcal{M})}{r})^d \cdot (\exp\{-s(\mathcal{L}_X(F_{f,\mathcal{D}}(\mathbf{z})) + \frac{\epsilon}{2})\}$$
$$+ \exp\{-s(\mathcal{L}_X(F_{f,\mathcal{D}}(\mathbf{z})) - \frac{\epsilon}{2})\}) + (\frac{2r\sigma L_f}{\epsilon})^2$$
(43)

For the case of **F2/F3**, we can have: $p(\epsilon) = \exp(-s\mathcal{L}_{X^+}(F_{f,\mathcal{D}}(\mathbf{z}) + \frac{\epsilon}{2}) + \exp(-s\mathcal{L}_{X^+}(F_{f,\mathcal{D}}(\mathbf{z}) - \frac{\epsilon}{2}) + \exp(-s\mathcal{L}_{X^-}(F_{f,\mathcal{D}}(\mathbf{z}) + \frac{\epsilon}{2}) + \exp(-s\mathcal{L}_{X^-}(F_{f,\mathcal{D}}(\mathbf{z}) - \frac{\epsilon}{2}))$. Then:

$$\mathbb{P}[\sup_{\mathbf{z}\in\mathcal{M}}|g(\mathbf{z})| \geq \epsilon] \leq (\frac{4\text{diam}(\mathcal{M})}{r})^d \cdot [\exp(-s\mathcal{L}_{X^+}(F_{f,\mathcal{D}}(\mathbf{z}) + \frac{\epsilon}{4}))$$
$$+ \exp(-s\mathcal{L}_{X^+}(F_{f,\mathcal{D}}(\mathbf{z}) - \frac{\epsilon}{4}))$$
$$+ \exp(-s\mathcal{L}_{X^-}(F_{f,\mathcal{D}}(\mathbf{z}) + \frac{\epsilon}{4}))$$
$$+ \exp(-s\mathcal{L}_{X^-}(F_{f,\mathcal{D}}(\mathbf{z}) - \frac{\epsilon}{4}))] + (\frac{2r\sigma L_f}{\epsilon})^2$$
(44)

Still, one can take $C = 2 \cdot 4^d$ here. In order to find smallest $s$ such that **F1/F2/F3** can satisfy the bound, we optimize for r and get the asymptotic value of the number of sample $s$ that provides $\epsilon$-accuracy(we assume $\mathcal{L}_X(F_{f,\mathcal{D}}(\mathbf{z}))$ or $\mathcal{L}_{X^{+/-}}(F_{f,\mathcal{D}}(\mathbf{z}))$ mentioned in Theorem 1 and Theorem 2 are constants:

$$s = \Theta(d\log(\frac{L_f\sigma(\text{diam}(\mathcal{M}))}{\epsilon}))$$
(45)

$\square$

## 8.7 On the Uniform Convergence of OMCs for Improving OMC Kernel Ridge Regression Guarantees

Recalling the setting in Theorem 2 of [11]:

**Theorem 4.** *Assume that a dataset $\mathcal{X} = \{x_1, x_2, ..., x_n\}$ is taken from a ball $\mathcal{B}$ of a fixed radius $r$ which is independent to the dimensionality of the data $n$, and size of dataset $N$, and the center $x_0$.*

*Consider kernel ridge regression adopting a smooth RBF kernel, especially Gaussian kernel. Let $\widehat{\Delta}_{iid}$ denote the smallest positive number such that $\widehat{\mathbf{K}}_{iid} + \lambda N\mathbf{I}_N$ is a $\Delta$-approximation of $\mathbf{K} + \lambda N\mathbf{I}_N$, where $\widehat{\mathbf{K}}_{iid}$ is an approximate kernel matrix obtained by using unstructured random features. Then for any $a > 0$,*

$$\mathbb{P}[\widehat{\Delta}_{iid} > a] \leq p_{N,m}^{iid}(\frac{a\sigma_{min}}{N}),$$
(46)

*where $p_{N,m}^{iid} = N^2 e^{-Cmx^2}$ for some universal constant $C > 0$, $m$ is the number of random features used, $\sigma_{min}$ is the smallest singular value of $\widehat{\mathbf{K}} + \lambda N\mathbf{I}_N$ and $N$ is the dataset size. If instead orthogonal random features are used then for the corresponding spectral parameter $\widehat{\Delta}_{ort}$ the following holds:*

$$\mathbb{P}[\widehat{\Delta}_{ort} > a] \leq p_{N,m}^{ort}(\frac{a\sigma_{min}}{N}),$$
(47)

*where function $p_{N,m}^{ort}$ satisfies: $p_{N,m}^{ort} < p_{N,m}^{iid}$, for n large enough.*

Based on this original version, we would like to offer a refined version as the following:

Rather than having $p_{N,m}^{iid} = N^2 e^{-Cmx^2}$, we can further remove $N^2$ by exploiting uniform convergence property if $z = x_i - x_j$ is in a compact set, $x_i, x_j$ are arbitrary two datapoints in the dataset, meaning that

$$p_{N,m}^{iid} = e^{-Cmx^2} \tag{48}$$

Following the same logic, we can still have $p_{N,m}^{ort} < p_{N,m}^{iid}$, for n large enough, resulting in a much stronger guarantee for kernel ridge regression. Proof is the following:

*Proof.* Motivated by [11], we can even substantially improve theoretical guarantees offered in its Theorem 2 with the uniform convergence property that we derived above. In order to achieve it, we will improve the Lemma 1 of [11]. We discuss all steps in detail below.

For an RBF kernel $\mathbf{K} : \mathbb{R}^n \times \mathbb{R}^n$, with a corresponding random feature map: $\Phi_{m,n} : \mathbb{R}^n \to \mathbb{R}^{2m}$, we can approximate it with a randomized kernel estimator $\widehat{\mathbf{K}}$. Assume that for any $i, j \in [N]$, the following holds for any $c > 0 : \mathbb{P}[|\Phi_{m,n}(x_i)^T \Phi_{m,n}(x_j) - \mathbf{K}(x_i, x_j)| > c] \leq g(c)$ for some fixed function $g : \mathbb{R} \to \mathbb{R}$. Then with probability at least $1 - g(c)$, matrix $\widehat{\mathbf{K}} + \lambda \mathbf{I}_N$ is a $\Delta$-spectral approximation of matrix $\mathbf{K} + \lambda \mathbf{I}_N$ for $\Delta = \frac{Nc}{\sigma_{min}}$, where $\sigma_{min}$ stands for the minimal singular value of $\mathbf{K} + \lambda \mathbf{I}_N$.

Denote $\mathbf{K} + \lambda N \mathbf{I}_N = \mathbf{V}^T \mathbf{\Sigma}^2 \mathbf{V}$, where an orthogonal matrix $\mathbf{V} \in \mathbb{R}^{N \times N}$ and a diagonal matrix $\mathbf{\Sigma} \in \mathbf{R}^{N \times N}$ define the eigendecomposition of $\mathbf{K} + \lambda N \mathbf{I}_N$. As shown in the paper, in order to prove that $\widehat{\mathbf{K}} + \lambda N \mathbf{I}_N$ is a $\Delta$-spectral approximation of $\mathbf{K} + \lambda N \mathbf{I}_N$, it suffices to show that:

$$||\mathbf{\Sigma}^{-1} \mathbf{V} \widehat{\mathbf{K}} \mathbf{V}^T \mathbf{\Sigma}^{-1} - \mathbf{\Sigma}^{-1} \mathbf{V} \mathbf{K} \mathbf{V}^T \mathbf{\Sigma}^{-1}||_2 \leq \Delta \tag{49}$$

With the definition of $l_2$ norm and Frobenius norm, we can have:

$$\mathbb{P}[||\mathbf{\Sigma}^{-1} \mathbf{V} \widehat{\mathbf{K}} \mathbf{V}^T \mathbf{\Sigma}^{-1} - \mathbf{\Sigma}^{-1} \mathbf{V} \mathbf{K} \mathbf{V}^T \mathbf{\Sigma}^{-1}||_2 > \Delta]$$
$$\leq \mathbb{P}[||\mathbf{\Sigma}^{-1} \mathbf{V}|| \widehat{\mathbf{K}} - \mathbf{K}||_F \mathbf{V}^T \mathbf{\Sigma}^{-1}||_2 > \Delta]$$
$$= \mathbb{P}[||\widehat{\mathbf{K}} - \mathbf{K}||_F^2 > \frac{\Delta^2}{||\mathbf{\Sigma}^{-1} \mathbf{V}||_2^2 \cdot ||\mathbf{V}^T \mathbf{\Sigma}^{-1}||_2^2}]$$
$$\leq \mathbb{P}[||\widehat{\mathbf{K}} - \mathbf{K}||_F^2 > \Delta^2 \sigma_{min}^2]. \tag{50}$$

The last inequality we use the fact that $||\mathbf{\Sigma}^{-1} \mathbf{V}||_2^2 \leq \frac{1}{\sigma_{min}}$ and $||\mathbf{V}^T \mathbf{\Sigma}^{-1}||_2^2 \leq \frac{1}{\sigma_{min}}$ because $\mathbf{V}$ is an isometric matrix.
Most importantly, we can refine the proof of lemma 1 in [11] using the uniform convergence property, provided that $z = x_i - x_j$ is in a compact set. Then the following inequalities hold:

$$\mathbb{P}[||\widehat{\mathbf{K}} - \mathbf{K}||_F^2 > \frac{\Delta^2}{||\mathbf{\Sigma}^{-1} \mathbf{V}||_2^2 \cdot ||\mathbf{V}^T \mathbf{\Sigma}^{-1}||_2^2}]$$
$$\leq \mathbb{P}[|\widehat{\mathbf{K}}_{i,j} - \mathbf{K}_{i,j}| > \frac{\Delta \sigma_{min}}{N}]$$
$$= \mathbb{P}[|\Phi_{m,n}(x_i)^T \Phi_{m,n}(x_j) - \mathbf{K}_{i,j}| > \frac{\Delta \sigma_{min}}{N}] \tag{51}$$

Therefore, the probability that $\widehat{\mathbf{K}} + \lambda \mathbf{I}_N$ is a $\Delta$-spectral approximation of $\mathbf{K} + \lambda \mathbf{I}_N$ is at least $1 - g(c)$ for $c = \frac{\Delta \sigma_{min}}{N}$. Afterwards, we can prove lemma 2,3,4,5 in [11] in the same way. Therefore, it shows that we can have a stronger concentration result for kernel ridge regression. $\square$

## APPENDIX B: Demystifying Orthogonal Monte Carlo and Beyond - Experiments

In our experiment with the particle algorithm (opt-NOMC), we use: $\eta = 1.0, \delta = 0.1, T = 50000$.

## 8.8 Clock Time Comparison for NOMC

In order to present the efficiency of our NOMC optimization procedure, we run our algorithm on a **single 6-core computer with Intel Core i7 CPU**, and parameter $d$ range from 8 to 256. The algorithm here uses the plain gradient descent method for optimization. Please note that this wall clock time is just a **one-time cost**, even if a new ensemble of samples is required at each iteration of the higher-level algorithm. In such a case that one-time optimized ensemble is simply randomly rotated using independently chosen random rotations, as mentioned in main text. Furthermore, we can always improve the efficiency by multi-machine parallelization, which however is not the focus of this work.

Figure 4: Clock time comparisons with $d = 8, 16, 32, 64, 128, 256$, and $s = 5d$. $D_{\max}$ is the maximum distance among all the points on the unit-sphere, and $D_{\min}$ represents the minimum of them. We use the difference between $D_{\max}$ and $D_{\min}$ as the y-axis, which should gradually decrease with each iterations. Besides, the red point in each line represents the first position where the absolute change in y-axis within the past 5000 iterations is below 0.01. We set parameters $\delta = 0.1$ and $\eta = 1$ in Algorithm 1.

| $d$ | 8 | 16 | 32 | 64 | 128 | 256 |
|---|---|---|---|---|---|---|
| Clock Time | 20 seconds | 2 minutes | 5 minutes | 22 minutes | 2 hours | 14 hours |

Table 8.8: Clock time comparison for different $d$. The time here represents the first time when the absolute change in $D_{\max} - D_{\min}$ within the past 5000 iterations is below 0.01 (same as the red point in Fig. 4).

## 8.9 Experimental Details for Kernel Approximation Experiment

In Sec.5, we present a result showing that our NOMC method indeed outperforms other algorithms. Specifically, we adopt mean squared error (MSE) as the error measure for pointwise estimation. As for the data set, rather than using theoretically simulated data, we adopted a variety of the data set from the UCI Machine Learning Repository for our experiments. Due to space constraints, we only select one of the experimental results from those data set, which is Letter Recognition Data Set. Also, this is one of the most popular and classical experimental data set. In our experiment, we compared 8 different kernels, which is shown in the table 6.7 below. For each kernel, we tested the performance of MC, QMC, B-OMC, NOMC for 10 multipliers, ranging from 1 to 10. For each multiplier, we performed 450 pointwise estimations to 100 randomly sampled data pairs and calculated the average of the MSE, in order to relieve the impact of single selection bias. Empirically, for the purpose of ensuring the kernel values in an appropriate range, we scaled the dataset using the mean distance of the 50th $l_2$ nearest neighbor for 1000 sampled datapoints [43].

This experiment is implemented in `Python 3.7` and executed on a standard `1.7 GHz Dual-Core Intel Core i7`.

| Kernel name | Kernel function | Function $\phi$ | Fourier density |
|---|---|---|---|
| Gaussian | $\sigma^2 \exp\left(-\frac{1}{2\lambda^2}\right)z^2$ | $\cos(\omega^T \mathbf{x} + b)$ | $\frac{\sigma^2}{(2\pi\lambda^2)^{n/2}} \exp -\frac{1}{2\lambda^2}\|w\|_2^2$ |
| Matérn[15] | $\sigma^2 \frac{2^{1-\nu}}{\Gamma(\nu)}(\sqrt{2\nu}z)^\nu K_\nu(\sqrt{2\nu}z)$ | $\cos(\omega^T \mathbf{x} + b)$ | $\frac{\Gamma(\nu+n/2)}{\Gamma(\nu)(2\nu\pi)^{n/2}}\left(1 + \frac{\|w\|^2}{2\nu}\right)^{-\nu-p/2}$ |
| Cauchy[34] | $\Pi_d \frac{2}{1+z_d^2}$ | $\cos(\omega^T \mathbf{x} + b)$ | $e^{-\|\omega\|_1}$ |
| Angular | $1 - \frac{2\theta_{\mathbf{x},\mathbf{y}}}{\pi}$ | $\text{sgn}(\omega^T x)$ | N/A |
| Quadratic | $\mathbb{E}_\omega[\phi(\mathbf{x})\phi(\mathbf{y})]$ | $(\omega^T x)^2$ | N/A |
| Tanh | $\mathbb{E}_\omega[\phi(\mathbf{x})\phi(\mathbf{y})]$ | $\tanh(\omega^T x)$ | N/A |
| Sine | $\mathbb{E}_\omega[\phi(\mathbf{x})\phi(\mathbf{y})]$ | $\sin(\omega^T x)$ | N/A |

Table 8.9 : Tested kernels, their corresponding kernel functions (we give compact form if it exists), mappings $\phi$ such that $K(\mathbf{x}, \mathbf{y}) = \mathbb{E}_\omega[\phi(\mathbf{x})\phi(\mathbf{y})]$ (used in MC sampling), and Fourier denssities (valid only for hift-invariant kernels). For Matérn kernel, $\Gamma(\cdot)$ denotes the gamma function, $K_\nu(\cdot)$ denotes the modified Bessel function of the second kind, and $\nu$ is a non-negative parameter. Parameter $\lambda$ denotes standard deviation, $\mathbf{z} = (z_1, ..., z_d)^\top = \mathbf{x} - \mathbf{y}$, $z = \|\mathbf{z}\|_2$ and $b \sim \text{Unif}[0, 2\pi]$.

## 8.10  Experimental Details for Sliced Wasserstein Distance Experiment

We run these experiments on a single 6-core computer with Intel Core i7 CPU. For the Sliced Wasserstein Distance experiments in Section 5, we use the same procedure as in the kernel approximation experiments and tested on 8 classes of distributions. For each class, we have two multivariate distributions with different means and covariance matrices. Following the formula in Eq. 10, we replaced the iid samples $\mathbf{u} \sim \text{Unif}(\mathcal{S}^{d-1})$ (which is the plain MC method) with samples from multiple orthogonal blocks, near orthogonal algorithms, and Halton sequences (which are B-OMC, NOMC and QMC respectively). We independently sample 100 thousands data points from each of the two distributions from the same class, and then compute the projections on the directions of $\mathbf{u}$. The specific details regarding mean and covariance matrix of each distribution are in Table 8.10. Let $\mathbf{A}$ be a $d \times d$ matrix with each entry generated from standard univariate gaussian distribution. Also let $\mathbf{D}$ be a $d \times d$ matrix obtained from the distribution of $\mathbf{A}$ by zeroing all off-diagonal values to zero. We take $\mathbf{M} \overset{\text{def}}{=} \mathbf{A}^\top \mathbf{A}/\sqrt{d}$ (note that $\mathbf{A}$ is positive semi-definite).

| Distribution name | Mean | Covariance Matrix | Parameter |
|---|---|---|---|
| Multivariate Gaussian | $(0, 0, ..., 0), (1, 1, ..., 1)$ | $\mathbf{M}_1, \mathbf{M}_2$ | N/A |
| Multivariate T | $(0, 0, ..., 0), (1, 1, ..., 1)$ | $\mathbf{M}_1, \mathbf{M}_2$ | df=10 |
| Multivariate Cauchy | $(0, 0, ..., 0), (1, 1, ..., 1)$ | $\mathbf{M}_1, \mathbf{M}_2$ | N/A |
| Multivariate Laplace | $(0, 0, ..., 0), (0, 0, ..., 0)$ | $\mathbf{M}_1, \mathbf{M}_2$ | N/A |
| Gaussian Mixture Q=2 | $(0, ..., 0, 1, ..., 1), (1, ..., 1, 0, ..., 0)$ | $\mathbf{D}_1, \mathbf{D}_2$ | N/A |
| Gaussian Mixture Q=3 | $(1, 1, 1, 1, 0, ..., 0), (0, ..., 0, 1, 1, 1)$ $(0, ..., 0, 1, 1, 1, 0, ..., 0)$ | $\mathbf{D}_1, \mathbf{D}_2, \mathbf{D}_3$ | N/A |
| Gaussian Mixture Q=4 | $(1, 1, 1, 1, 0, ..., 0), (0, 0, 1, 1, 0, ..., 0)$ $(0, ..., 0, 1, 1, 0, ..., 0), (0, ..., 0, 1, 1)$ | $\mathbf{D}_1, \mathbf{D}_2, \mathbf{D}_3, \mathbf{D}_4$ | N/A |
| Inverse Wishart | $(0, 0, ..., 0), (1, 1, ..., 1)$ | $\mathbf{M}_1, \mathbf{M}_2$ | $\nu = 10$ |

Table 8.10 : Tested classes of distributions (from each we sampled two distributions for SWD computations), their corresponding means of modes, covariance matrices for different modes and other parameters (if applicable).