[Reviews · NeurIPS 2020]

Review 1

Summary and Contributions: In this work, the authors provide a simple technique based on negative dependence to prove the exponential bounds of Orthogonal Monte Carlo (OMC) under the monotone assumption of the integrand function. The bound of approximation error is tighter than Monte Carlo sampling. Moreover, the authors prove a uniform convergence result of OMC. Although server kind of error bounds have studied in the literature, the technique in this paper seems simple. And it results in some tighter bounds.

Strengths: 1. The proof technique based on negative dependence and can covers many bounds as shown in the paper. 2. It seems that tighter bound is achieved for PNG and Matern kernel with any \nu.

Weaknesses: 1. I do have concerns about the Near-Orthogonal Monte Carlo part in section 4. Several closely related works are missing. In [2], the author also generates samples on the sphere by minimizing energy for kernel approximation. As shown in a recent survey for kernel approximation [1], the method in [2] (SSF) is still a strong baseline. It is better to have a discussion about the relation to SSF. 2. The proposed method in section 4.2 is a concatenation of real and imaginary part of the construction in [3]. In addition, Lemm3 in section 4.2 is a direct result of Lemma 2.2 in [3]. 3. At line 596 in the appendix, the constant is defined as maximum of the norm of the gradient. Does the assumption changed to function g is Lg-Lipschitz smooth? Why need to consider the gradient here? [1] Liu et al. Random Features for Kernel Approximation: A Survey on Algorithms, Theory, and Beyond. [2] Lyu . Spherical structured feature maps for kernel approximation [3] Xu. Deterministic Sampling of Sparse Trigonometric Polynomials.

Correctness: The theoretical part does not have fatal errors after a rough check.

Clarity: The paper is well written.

Relation to Prior Work: The relation to the method in [3] and [2] needs a discussion. 1. The proposed method in section 4.2 is a concatenation of real and imaginary part of the construction in [3]. In addition, Lemm3 in section 4.2 is a direct result of Lemma 2.2 in [3]. The paper can be improved by a discussion about the connection. 2. In [2], the author also generates samples on the sphere by minimizing energy for kernel approximation. It is better to include a discussion about the relation between the proposed method and [2].

Reproducibility: Yes

Additional Feedback: The paper can be improved by a clear discussion about relation to the existing work. ====================================POST REBUTTAL ============== Thanks to the authors' responses. I agree the main contribution of this work is the theoretical results of OMCs. I think the NOMC method (sec.4) is a bit separable from the theory. The novelty of the NOMC is limited. It is better for the authors to revise the contribution part (L84-L87) and section 4, and highlight the main theoretical contribution of OMCs.


Review 2

Summary and Contributions: This paper analyzes the orthogonal Monte-Carlo algorithm using negatively dependent random variables. The authors obtain results that provide bounds on the error probabilities. I have the following questions on the paper: 1. In the definition of the function f_Z in section 2, where Z is an ordered subset of P(R^d), I believe P(R^d) refers to the set of probability measures on R^d. If not, please specify what it is. Give an example of the ordered set, i.e., how do you order probability measures? Also, do you associate a metric topology with P(R^d) and would you not need properties such as completeness and separability of the space of probability measures w.r.t. this metric? 2. In F2, page 4, how are f+ and f- defined? What do the + and - signs represent? Give an example and is the decomposition of f into f+ and f- unique? For instance, wouldn't f+ and f- be both nonnegative functions? 3. In the remark following F2, you have said what even[f] is but again is there a procedure for finding even[f]+ and even[f]-? If there is no unique way in which these quantities are defined, at least an example must be provided to clarify what these could potentially be. 4. The claim concerning yours is the first work that addresses discontinuous functions: I don't see the function properties well specified here. For instance, wouldn't you need the functions to be at least measurable? 5. Theorem 1: What are points a and b and how are they specified, see p(epsilon) in the second line below Equation (5). Also, then does it mean that p(epsilon) defined one line below (5) is defined outside the interval [a,b]? It is not clear what these points a and b are and how are these obtained in practice? 6.

Strengths: The strength of this paper is an analysis of the OMC algorithm.

Weaknesses: Several things are not very clear and are detailed above. It would make better sense if these things are clarified in the paper and an illustrative example provided.

Correctness: The claims appear correct though I didn't check them fully. A few empirical experiments are also shown.

Clarity: The paper is not entirely well written. There are places where more clarity is needed. Also, I found a few typos and incorrect sentence constructions. For example, "...in particular that OMCs uniformly convergence for all RBF kernels:"

Relation to Prior Work: Relation with prior work is reasonably clarified.

Reproducibility: Yes

Additional Feedback: Post rebuttal: The authors have said they will clarify many of the issues raised in the revision. I will look forward to an improved paper that addresses the concerns raised.


Review 3

Summary and Contributions: The paper analyzes the orthogonal Monte Carlo (OMC) method, showing that they perform at least as well as iid sampling. By using the theory of negatively dependent random variables, the paper provides high probability bound on the error of OMC, which leads to uniform convergence results of OMC in kernel ridge regression. The paper also proposes a new sampling method called near-orthogonal Monte Carlo (NOMC) that improves on the common construction of using blocks of OMC. The paper empirically validates this claim on kernel approximation and sliced Wasserstein distance approximation tasks.

Strengths: 1. Novel theoretical results on OMC, a common sampling method in many applications (random projection, kernel approximation, etc.). This provides much needed theoretical guarantees on this method. The insight of using negatively dependent random variables is simple and effective. I enjoyed reading Section 3 in particular. 2. The proposed near-orthogonal Monte Carlo method outperforms block construction of OMC, as validated on the kernel approximation task and sliced Wasserstein disctance approximation task. Though the results are on synthetic tasks and not on downstream applications, this is impressive as OMC and block OMC commonly performs well on these tasks.

Weaknesses: 1. Unclear connection between the theory of orthogonal Monte Carlo (section 3) and the proposed method (near-orthogonal Monte Carlo, section 4-5). To me, the proposed method seems unrelated to the theory. The experiment in section 5 doesn't seem to connect to the theory either. Perhaps the authors can expand on the connection between the two parts of the paper. 2. Lack of details/reproducibility. The algorithm alg-NOMC in Section 4.2 (which is also the method used in the experiments in section 5) is not described in details. The hyperparameter tuning procedure (e.g. delta in the opt-NOMC method) is not described. Without code, this might make it hard to reproduce the results. In particular, it's not clear how to construct s random omegas vectors in section 4.2, which is needed to implement the algorithm. 3. Lack of experiment on the downstream applications. It would be interesting to see if NOMC improves in the downstream applications mentioend, such as in kernel regression and generative models.

Correctness: The theoretical results seem correct, though I have not carefully checked all the proofs in the appendix. The empirical methodology is correct. Lemma 3 in Section 4.2 does not seem to have a proof in the appendix. The citation [38] doesn't seem to have a proof of the lemma either (at least not in the form stated).

Clarity: The paper is sufficiently clear, though section 4.2 in particular could benefit from better description of the algorithm. The authors could also include in the appendix derivations to show how different classes of functions (e.g. JLT, RBF) satisfy the conditions F1-F3 (section 3).

Relation to Prior Work: Discussion on differences from previous work is sufficient.

Reproducibility: No

Additional Feedback: Question: line 144 claims "It is a standard fact from probability theory that L_X(a) > 0". Is there a reference for this claim? ===== Update after authors' rebuttal: Thanks to the authors for explaining the construction of alg-NOMC and the connection between the negative dependence theory nad the proposed algorithms (NOMC). Based on the downstream result on GP regression, it does seem that the proposed NOMC improves on block-OMC construction. However, as pointed out by other reviews, the paper could benefit from clearer presentation that emphasizes the main contribution (the theoretical guarantees of OMC).


Review 4

Summary and Contributions: This paper contributes to the theoretical understanding of the Orthogonal Monte Carlo method, with new concentration results supporting the empirical behavior shown in previous work. The key ingredient is the non negative dependence property satisfied by the random variables |x_i^t z| where x_i belongs to an orthogonal ensemble (Lemma 1). Several implications on a more applied side, including kernel ridge regression with OMC, are presented in appendix. A numerical method is proposed to generate a nearly orthogonal design on the hypersphere which contains more points than the ambient dimension.

Strengths: The theoretical derivations presented in the paper seem novel and valuable for the community, they support the use of negatively correlated samples for structured Monte Carlo methods.

Weaknesses: The paper is not easy to read, the presentation of the results and their originality is sometimes not very clear. The idea underlying NOMC: minimizing an energy function on the hyper-sphere is already present in the literature.

Correctness: - l 231 "We construct NOMC-samples to make angles between any two samples close to orthogonal"to make angles between any two samples close to orthogonal (...)" This claim might be misleading especially when looking at Figure 1. - Both the MSE notation and the notion of isotropic distribution are crucial but remain undefined. - l 252 "WLOG we can assume [...] D is a uniform distribution on the sphere Unif(Sd 1), since for other isotropic [...] only need to conduct later cheap renormalization of samples’ lengths." Can you give more details on that claim and further references ? - l 84 "We propose two new paradigms for constructing structured samples" I can only find one algorithm opt-NOMC, while Section 4.2 "alg-NOMC" does not seem to present a second method but rather invokes some non-trivial results from algebraic geometry in a very unclear way. - The time complexity of the algorithm is not presented. Apart from the MSE performance, the CPU time and complexity comparison with existing methods are also crucial, especially for practical purposes. What is the cost of the optimization procedure ? What is the complexity of rotating the points

Clarity: In its current form, the paper is not easy to read, I find it very dense, with a lot of notation and some inconsistencies. The writing of some sections of the paper feels unfinished (see above). A less dense paper with a cleaner exposition of the theoretical results only may certainly serve as a milestone for further research on OMC.

Relation to Prior Work: - It is not clear whether Lemma 1 and 2 are new or already derived in previous work. - It seems that the algorithmic part of this work uses longstanding ideas in mathematics which are not mentioned nor compared to. See e.g. "Distributing many points on a sphere" Saff and Kuijlaars "Distributing many points on spheres: Minimal energy and designs" Brauchart and Grabner "Good permutations for deterministic scrambled Halton sequences in terms of -discrepancy" Vandewoestyne and Cools

Reproducibility: Yes

Additional Feedback: - What do the colors of Figure 1 refer to? - How to adapt OMC to non-isotropic distributions, when the support of integration is the unit cube for instance ? Typos / inconsistencies in the notation - Sec. / Section - Eq. / Equation - missing punctuation at the end of some equations - "probabilistic distribution" -> probabilitY distribution - Legendre symbol already has a meaning in number theory and the terminology is not reused later in the paper - l 252 "WLOG" - a lot of ":", see, e.g., l 159 ": ... :" - l 160 transpose symbol is missing - l 489 "which can be rewritTEN" - please avoid extra () - l 37 ([15]), l293 "(QMC)([7])" l 688 ".(see:[40])" l 227 "(if all points are in the compact set) (Appendix: Sec.7.7)" - the methods are not ordered in the same way in Figures 2 and 3 - extra expectation symbols in (37) Suggestions - The notation "s" for the number of samples is not common compared to "n" - The subscripts F_{f,\calD} may be dropped for readability - Size adaptive [ { ( in the equations could ease readability of the equations - Legend Figure 1 "We see that NOMC produces most uniformly distributed samples." homogeneously might preferred to uniformly. **EDIT** While the theoretical contribution is indeed valuable, the rebuttal and the discussions did not resolve my concerns about the structure of the paper (only two pages for the main contributions), the clarity of the exposition and comparison with existing literature, hence an overall score of 5.

[Author Response · NeurIPS 2020]

We would like to sincerely thank all the reviewers for reading the paper carefully and their very valuable feedback.

**General comments:**

**Presentation:** For the camera-ready, we will de-densify the main body of the paper to improve clarity (by moving technical results to the Appendix), fix all typos and incorporate all suggested notation improvements.

**Legendre symbol:** Current name is unfortunate (conflict with number theory version), thus we will change it. Inequality $\mathcal{L}_X(a) > 0$ follows directly from the properties of the Legendre **Transform** and we will clarify it in the final version.

**Reviewer 1:**

**Relation to previous work:** We will add detailed discussion regarding [1], [2] (and other works on energy-minimization techniques) as well as [3] in the final version. As opposed to [1] and [2]: we consider different energy function, evaluate on kernels beyond Gaussian and arc-cosine classes, target extra applications beyond kernels and propose simpler OPT-NOMC algorithm. OPT-NOMC is not our main contribution, is not even our only NOMC version. ALG-NOMC is unrelated to energy-based methods. Lemma 3 follows from Weil's results (explicitly stated in the text) thus it is very interesting, but not surprising that it implied also by [3] which exploits them too, yet in a very different setting (recovering sparse multivariate trigonometric polynomials). Finally, **the main contribution of this work are our theoretical results on ND which preceed Sec. 4**. **L.596:** This is for continuous gradient since $\mathcal{M}$ is compact. General cases are handled by finite difference approach. We will clarify it in the final version (see also: comment in l.598-599.)

**Reviewer 2:**

**Definition of $f_{\mathcal{Z}}$:** $\mathcal{Z}$ refers to ordered subset of $\mathbb{R}^d$, thus we should have: $\mathcal{Z} \subseteq \mathbb{R}^d$, thank you for catching this typo.

**Functions $f^{+/-}$ / even$[f]^{+/-}$:** $+/-$ corresponds to "increasing/decreasing" in $|u|$". Functions $f^{+/-}$ generally are not uniquely defined, but in most applications will simply relate to positive/negative part of the Taylor series (TS) for $f$ (if TS exists, both parts are finite and TS does not contain odd-power terms). Prominent examples include Gaussian and Matern kernel, see: Table 1 where we also put formulae on $f^+$ and $f^-$. Here $f^{+/-}$ can be directly obtained from TS. Similar analysis is true for even$[f]^{+/-}$ (see: Table 1). Now we only need to filter out from TS odd-power terms.

**Discontinuous $f$:** we need measurability, but no other properties are required.

**Points $a$ and $b$:** **a** and **b**, these are simply: the lower and upper bound on $f$. The second formula for $p(\epsilon)$ is for the case of bounded $f$ and the first one is for the case of unbounded $f$. **We will clarify all these in the final version.**

**Reviewer 3:**

**Connection btw theory and NOMC:** Thank you for the comment. OPT/ALG-NOMC algorithms were inspired by negative dependence results since some of the classic examples of negatively-dependent systems come from energy-based configurations (statistical Physics) and algebraic theory. We will properly describe this link in the final version.

**ALG-NOMC:** Eq. 8-9 uniquely define the ensemble of samples and consequently, completely determine the algorithm. Vectors from $\Omega$ are defined as: $(a_1, b_1, ..., a_p, b_p)$ (see: Eq. 9), where $a_j, b_j$ are: real and imaginary part of $g_{c_1,...,c_r}(j-1)$ (see: Eq, 8 for the definition of $g_{c_1,...,c_r}$). Different vectors correspond to different $(c_1, ..., c_r) \in (\mathbb{F}_p)^r$. This compact construction does not require any optimization, thus it was not presented originally in the separate algorithmic box. However we do agree with the reviewer that, since the content is very technical, for the clarity of the exposition, it would benefit from separate algorithm box and more careful explanation. We will do it in the camera-ready version.

**Lemma 3 in Sec. 4.2:** The proof follows directly from the Weil conjecture and related construction is one of the flagship examples from algebraic number theory of the close-to-optimal ensemble (with respect to size, which is what Lemma 3 says) with Kabatjanskii-Levenstein Lemma establishing tight upper bound. In the camera-ready, we will clarify it and add all details since Weil conjecture and its implications deserve a separate paragraph.

**Functions satisfying properties:** $F1$-$F3$: Table 1 provides several examples of functions from classes $F1$-$F3$.

**Hyperparameters:** We chose $\eta = 1.0, \delta = 0.1, T = 50000$ (thus did not tune them) as stated at the beginning of Appendix B. In the main body in l.294 we explain that: "Additional experimental details are in the Appendix."

**Downstream experiments:** We run approximate GP regression from "*The Geometry of Random Features*" (Fig.8) for Gaussian kernel and $s/d = 1, 2, 3, 4$, obtaining: OPT-NOMC: **0.5**, **0.4**, **0.36**, **0.31**, BEST: **0.54**, **0.44**, **0.43**, **0.39** (average test RMSE), where BEST is the best result across methods considered there (will be added to final version).

**Reviewer 4:**

**Isotropic distribution:** An isotropic distribution $\mathcal{D}_{\text{iso}} \in \mathcal{P}(\mathbb{R}^d)$ is defined as having pdf constant on every sphere centered at 0. We will explicitly state it in the final version. Thus, by definition, if $\mathbf{v} \sim \mathcal{D}_{\text{iso}}$, then $\mathbf{v}$ can be rewritten as: $v = r * \mathbf{u}$, where $\mathbf{u} \sim \text{Unif}(\mathcal{S}^{d-1})$ and $r$ is an independent random variable defining how to renormalize length of $\mathbf{u}$ to get $\mathbf{v}$ (see also: *The Geometry of Random Features*, AISTATS 2018).

**Time Complexity:** As explained in l.261-265, apart from one-time extra cost of the optimization (and only for OPT-NOMC method, see: Sec. 7.8, where we run detailed ablation studies over $d$ for that wall clock time), time complexity is **the same**. Cost of a true random rotation is cubic in $d$, but regular OMC method also applies random rotations.

**Miscellaneous:** Lemma 1,2 are new. **Misleading claim near Fig. 1**: We will clarify that claim is for large $d$. **Non-isotropic distributions:** it is an interesting topic, but beyond the scope of this work since OMCs were designed for isotropic distributions. **Colors in Fig. 1** indicate direction of the vector (red: head of the vector, blue: tail). **MSE:** The MSE (mean squared error) is defined where we use it the first time, i.e. at the beginning of Theorem 1. See also, our response to Reviewer 1 regarding relation to **previous work** (minimizing an energy function on the hyper-sphere).

[Meta-Review · NeurIPS 2020]

The work presents theory behind the behavior of OMC methods based on negatively dependent random variables, which yields tighter bounds than MC, and the work also establishes uniform convergence bounds. The reviewers appreciated the theoretical development for OMC, the results appear new and carefully done. The work also proposed NOMC which has empirically promising results. Reviewers have concerns about the proposed NOMC, including missed connections with closely related literature, unclear connections with the theory of OMC established in the paper, no evidence of empirical evidence in downstream applications. The author response addressed some concerns, but some concerns persisted. The work will be strengthened by addressing these concerns.